# G-quadruplexes originating from evolutionary conserved L1 elements interfere with neuronal gene expression in Alzheimer's disease

Roy Hanna[1], Anthony Flamier[1,3], Andrea Barabino[1] & Gilbert Bernier [1,2✉]

DNA sequences containing consecutive guanines organized in 4-interspaced tandem repeats can form stable single-stranded secondary structures, called G-quadruplexes (G4). Herein, we report that the Polycomb group protein BMI1 is enriched at heterochromatin regions containing putative G4 DNA sequences, and that G4 structures accumulate in cells with reduced BMI1 expression and/or relaxed chromatin, including sporadic Alzheimer's disease (AD) neurons. In AD neurons, G4 structures preferentially accumulate in lamina-associated domains, and this is rescued by re-establishing chromatin compaction. ChIP-seq analyses reveal that G4 peaks correspond to evolutionary conserved Long Interspersed Element-1 (L1) sequences predicted to be transcriptionally active. Hence, G4 structures co-localize with RNAPII, and inhibition of transcription can reverse the G4 phenotype without affecting chromatin's state, thus uncoupling both components. Intragenic G4 structures affecting splicing events are furthermore associated with reduced neuronal gene expression in AD. Active L1 sequences are thus at the origin of most G4 structures observed in human neurons.

[1] Stem Cell and Developmental Biology Laboratory, Hôpital Maisonneuve-Rosemont, Montreal, QC, Canada. [2] Department of Neurosciences, University of Montreal, Montreal, QC, Canada. [3] Present address: Whitehead Institute of Biomedical Research, Cambridge, MA, USA. ✉email: gbernier.hmr@ssss.gouv.qc.ca

Nucleosomal histone proteins are regulated by post-translational modifications that can impact transcription, replication, and repair. BMI1 is a component of the Polycomb Repressive Complex 1 (PRC1). The PRC1 promotes chromatin compaction and gene silencing in part through its E3-mono-ubiquitin ligase activity mediated by RING1a/b on histone H2A at lysine 119 (H2A$^{ub}$)[1–3]. The PRC1 is recruited to facultative heterochromatin to maintain repression at developmental and senescence-associated genes[4–8]. BMI1 is also enriched at constitutive heterochromatin, where it co-purifies with ATRX, HP1, DEK1, and Lamins[9]. BMI1 inactivation in human dermal fibroblasts (HDFs) results in loss of heterochromatin and transcriptional de-repression of repetitive DNA sequences[9]. More recently, reduced neuronal expression of BMI1 was found associated with sporadic late-onset Alzheimer's disease (AD)[10]. Acute BMI1 inactivation in cultured human neurons can also recapitulate AD-associated hallmarks, including the accumulation of beta-amyloid and hyper-phosphorylated Tau[10]. Aged mice hemizygous for Bmi1 (Bmi1$^{+/-}$) also develop, along with some progeroid features, AD-like behavioral, and neuropathological phenotypes[11]. Furthermore, loss of heterochromatin and genomic instability at repetitive DNA sequences were described as new molecular characteristics present in cortical neurons from both Bmi1$^{+/-}$ mice and AD cases[11,12]. Loss of heterochromatin and transcriptional activation of specific classes of endogenous retroelements occur in neurodegenerative tauopathies and in animal models of Tau over-expression[13,14]. Notably, advanced aging is the greatest risk factor to develop AD[15,16], and many anomalies present in AD patient's neurons in situ, such as relaxed heterochromatin and nuclear envelope defects, are considered as hallmarks of aging[15,17–19]. Accordingly, it was proposed that AD may represent an acquired laminopathy[19,20].

Interestingly, most genetically inherited progeroid syndromes, such as Hutchinson–Gilford progeria, Werner, Bloom, and Xeroderma pigmentosum, present heterochromatin relaxation and genomic instability phenotypes[21–23]. With the exception of Hutchinson–Gilford progeria, which is linked to mutations in LMNA (encoding for the nuclear envelope protein Lamin A), the other progeroid disease genes encode DNA damage and/or repair proteins. More specifically, Werner (WRN), Xeroderma pigmentosum (XPB, XPD), and Bloom (BLM) gene products encode DNA helicases that can resolve G-quadruplex (G4) DNA's secondary structures (also called structured DNA or G-quadruplexes) stabilized by Hoogsteen hydrogen bonds between guanines (G)[24–26]. Notably, the DNA-dependent ATPase and helicase ATRX is enriched at repetitive DNA sequences predicted to form G4 structures, and ATRX can physically bind structured DNA in vitro[27]. Persistent G4 structures have been proposed to represent a threat to genomic stability and gene function by interfering with fork elongation and DNA repair during replication and transcription[28–31]. In normal physiological conditions, however, G4 structures may be necessary for the control of gene expression, maintenance of telomeres, and establishment of replication origins[32].

We report here that putative G4 DNA sequences are significantly enriched in BMI1 chromatin immunoprecipitation and sequencing (ChIP-seq) data sets, and that BMI1 deficiency in HDFs or neurons results in the induction of G4 structures. In AD neurons with relaxed heterochromatin, G4 structures accumulate at lamina-associated domains (LADs) and peri-nucleolar heterochromatin. Likewise, normal neurons exposed to drugs that open chromatin also present massive accumulation of G4 structures. ChIP-seq analyses of post-mitotic human neurons using the 1H6 antibody further show that G4 structures correspond to "active" evolutionary conserved Long Interspersed Element-1 (L1) sequences. Consistently, inhibition of RNA Polymerase II (RNAPII) can reverse the induction of G4 structures without altering the chromatin compaction state. Importantly, G4 structures present at specific loci in AD neurons are associated with reduced gene expression and perturbed alternative splicing. Chromatin-mediated transcriptional repression of L1 sequences thus prevents excessive formation of G4 structures in human neurons, which otherwise can interfere with gene expression.

## Results

Using public BMI1 ChIP-seq data sets, we annotated all BMI1-enriched chromatin regions and tested their propensity to form putative G4 DNA structures using the Quadparser algorithm[33]. Using four different sets of Quadparser parameters, we found that BMI1 was significantly enriched at chromatin regions predicted to form G4 structures independently of the Quadparser stringency (Fig. 1a). We further investigated whether BMI1 association with G4 motifs was comparable to the one observed in proteins known to physically interact with structured DNA such as the ATRX, XPB, and XPD helicases[25,27]. We annotated ATRX, XPB, and XPD ChIP-seq peaks alongside BMI1 peaks for the presence of putative G4. While ~45% of ATRX, XPB, and XPD peaks were associated with putative G4, only ~6% of BMI1 peaks showed the same association (Fig. 1b), suggesting that BMI1 does not bind G4 structures but is rather enriched at chromatin regions with the propensity to generate G4. We also analyzed the ChIP-seq peaks obtained with the BG4 antibody, which recognizes G4 structures. As reported, we found that ~60% of BG4 peaks contained a putative G4 DNA sequence (Fig. 1b)[34]. We next compared the putative G4 DNA sequences contained in the BG4, XPB, and the BMI1 peaks. Surprisingly, while ~30% of the putative G4 DNA sequences linked by BG4 were enriched for XPB, only 0.05% were enriched for BMI1 (Fig. 1c). Similarly, only 0.15% of the putative G4 sequences bound by XPB were enriched by BMI1 (Fig. 1c). Since BG4 peaks were reported to be predominant at the promoter of actively transcribed genes, these results suggested that BMI1 peaks containing putative G4 DNA sequences are rarely present in actively transcribed regions.

To test the possibility that BMI1 function is required to prevent the formation of G4 structures, we used the 1H6 and BG4 antibodies, which recognize G4 structures[34,35]. In early passage normal HDFs, we noticed that the baseline level of 1H6 and BG4 immunoreactivity was relatively low (Fig. 1d and Supplementary Fig. 1b). However, we observed a robust nuclear and modest cytoplasmic immunoreactivity for 1H6 and BG4 in BMI1-knockdown HDFs (shBMI1) (Fig. 1d and Supplementary Fig. 1b), suggesting accumulation of both G4 DNA and G4 RNA structures[36,37]. To test for the specificity of the signal obtained, we fixed naive cells with paraformaldehyde (PFA) and then exposed them to HCL, an agent that denatures DNA's secondary structure. HCL is commonly used in transmission electron microscopy and immunohistochemistry on paraffin sections. We found that HCL treatment resulted in non-specific immuno-labeling when compared to cells retaining a native chromatin state (Fig. 1d and Supplementary Fig. 1b). This revealed that G4-specific antibodies should only be used on cells with native chromatin. Pyridostatin is an agent that stabilizes spontaneously forming G4 structures, and we found that 1H6 immuno-labeling was significantly increased after exposing HDFs to pyridostatin (Fig. 1e)[38]. To further validate our observations, we performed colocalization studies with an antibody against the Werner (WRN) helicase. WRN is predicted to bind and unwind G4 structures[39–41]. In control HDFs, WRN and 1H6 levels were low and did not colocalize (Fig. 1f and Supplementary Fig. 2a). They were, however, significantly induced after BMI1 knockdown or after exposition to pyridostatin (Supplementary Fig. 2a). In both cases,

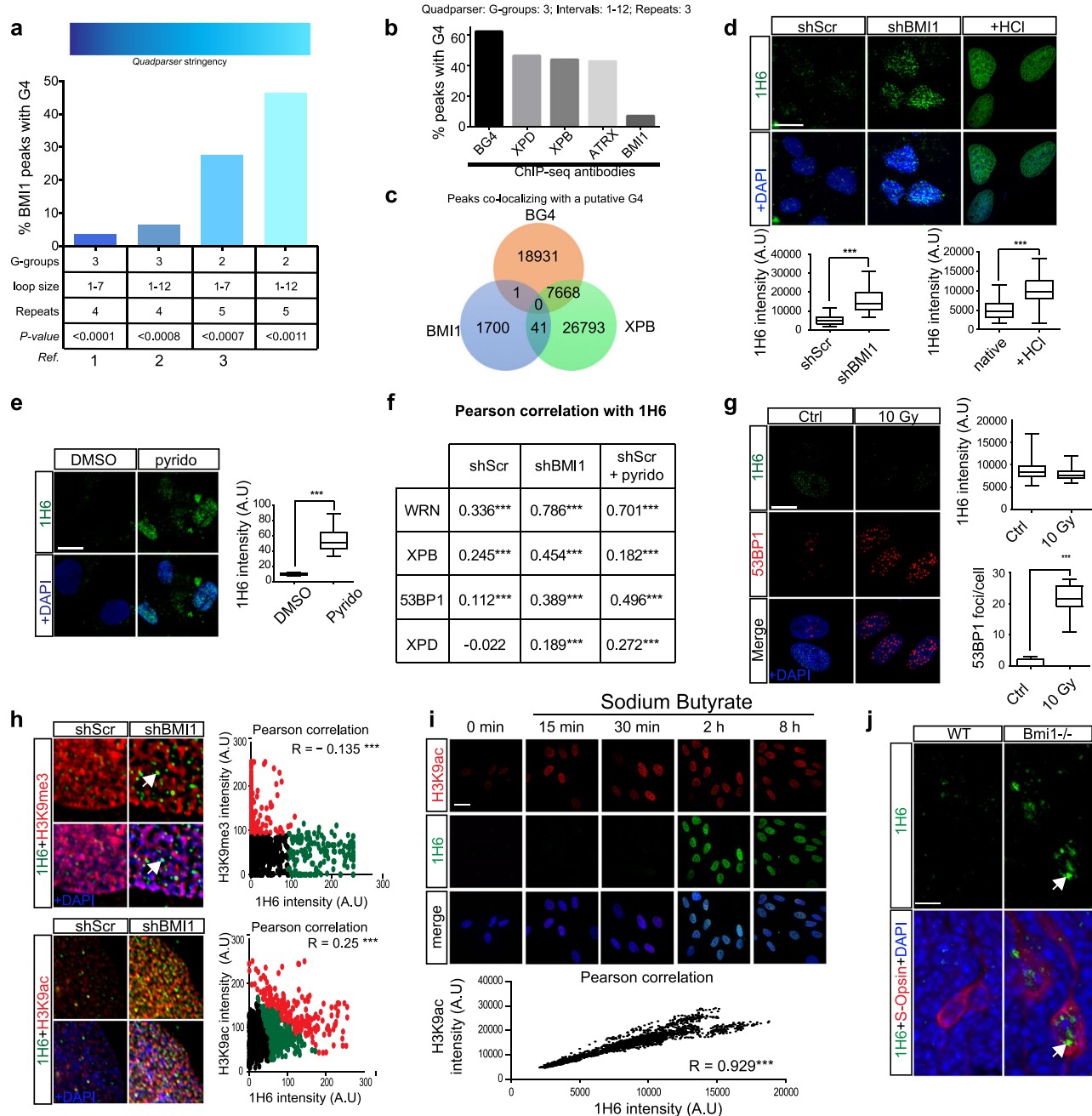

1H6 and WRN presented a very high coefficient of colocalization (Pearson correlation: 0.79 for sh*BMI1*; 0.70 for shScr + pyridostatin) (Fig. 1f). Relatively strong colocalization between 1H6 and XPB or 53BP1 (but not XPD) was also observed (Fig. 1f and Supplementary Fig. 2a). Importantly, G4 structures were not detected with 1H6 after treatment of HDFs with 10 Gy of gamma rays, which induce DNA double-strand breaks (Fig. 1g)[42]. To test if G4 structures were associated with the formation of DNA-RNA hybrids (also referred to as R-loops), we analyzed *BMI1*-knockdown HDFs using the S9.6 antibody[43,44]. We observed that in both control and *BMI1*-knockdown HDFs, the S9.6 signal was largely restricted to the nucleolus (Supplementary Fig. 1c) and that, while nucleoli were larger in *BMI1*-knockdown cells, no significant difference in S9.6 labeling was observed outside the nucleolus (Supplementary Fig. 1c, d). Importantly, while DNAse1 treatment could prevent 1H6 labeling in HDFs (Supplementary Fig. 1e), RNAse H and RNAse A were unable to do so

(Supplementary Fig. 1f), thus suggesting that nuclear structures recognized by the 1H6 antibody are DNA-based, and not RNA-DNA hybrids or secondary RNA structures. Taken together, these experiments suggest that nuclear foci visualized using the 1H6 antibody represent genuine G4 structures.

Given that BMI1 is enriched at heterochromatin and BMI1 deficiency results in loss of heterochromatin compaction[9], we reasoned that BMI1 activity might counteract the induction of G4 structures through chromatin compaction. The H3K9$^{me3}$ histone mark labels constitutive heterochromatin regions, which are in general transcriptionally inactive, while the H3K9$^{ac}$ histone mark labels "active" and open chromatin regions. Accordingly, reduced H3K9$^{me3}$ labeling was observed in HDFs 16 h after transfection with a *BMI1*-targeting shRNA vector, suggesting loss of heterochromatin (Fig. 1h and Supplementary Fig. 3a). Notably, nearly all 1H6 foci observed following *BMI1* knockdown did not colocalize with H3K9$^{me3}$ (Fig. 1h, Pearson coefficient correlation

**Fig. 1 Chromatin compaction prevents excessive formation of G4 structures. a** Shown is the proportion of BMI1 ChIP-seq peaks containing a putative G-quadruplex motif according to the Quadparser algorithm and using four independent sets of parameters. Top: color gradient indicating the stringency of the Quadparser parameters. G4, G-quadruplex; 1 Gray et al. 2014 parameters[96]; 2 Gray et al. 2014 and Zizza et al. 2016[97]; 3 Law et al. 2010[27]; *P* value: probability value based on generating six sets of 3542 randomly positioned probes and annotated for G4 motifs. **b** The proportion of ChIP-seq peaks for XPB, XPD, ATRX, and BMI1 containing a putative G-quadruplex motif according to the Quadparser algorithm. Gray et al. parameters were used to annotate all ChIP-seq data sets. G4, G-quadruplex. **c** Venn diagram for BMI1, XPB, and BG4 (G4-seq) ChIP-seq peaks colocalizing with a putative G-quadruplex according to Gray et al. Quadparser parameters. **d** Formaldehyde fixed HCA2 cells infected with shScramble or shBMI1 viruses, or denatured using 3 M of HCl, were immunolabeled and counterstained with DAPI and 1H6 antibody. These antibodies were used to detect G-quadruplexes structures. The graphs show the quantification of 1H6 signal intensity in each cell with the relevant unpaired *t*-test with two tails. $P \leq 0.05^*, \leq 0.01^{**}, \leq 0.001^{***}$. The box plots show the minima and the maxima values delimited by the whisper, the box delimites the 25th percentile and 75th percentile, and the line inside the box define the mean. A total of 100 cells were quantified in each condition. Scale bar: 10 μm. **e** Formaldehyde fixed HCA2, treated for 16 h with vehicle (DMSO) or 5 μM of pyridostatin (pyrido), were immunolabeled and counterstained with DAPI. The 1H6 antibody was used to detect G4 structures. Scale bar: 10 μm. The graphs show the quantification of 1H6 signal intensity in each cell from Fig. 1e with the relevant unpaired *t*-test with two tails. $P \leq 0.05^*, \leq 0.01^{**}, \leq 0.001^{***}$. The box plots show the minima and the maxima values delimited by the whisper, the box delimites the 25th percentile and 75th percentile, and the line inside the box define the mean. A total of 100 cells were quantified in each condition. **f** Pearson correlation between 1H6 and various helicases colocalization was calculated from the immunofluorescence pictures (see Supplementary Fig. 1c). **g** The 10 Gy irradiated HCA2 cells were fixed then immunolabeled and counterstained with DAPI. The graphs show the quantification of 1H6 and 53BP1 signal intensity in each cell. An average of 150 cells was analyzed per condition with the relevant *t*-test. $P \leq 0.05^*, \leq 0.01^{**}, \leq 0.001^{***}$. The box plots show the minima and the maxima values delimited by the whisper, the box delimites the 25th percentile and 75th percentile, and the line inside the box define the mean. Scale bar: 10 μm. **h** Formaldehyde fixed HCA2 cells infected with shScramble or shBMI1 viruses were co-immunolabeled with 1H6 and H3K9me3 or H3K9ac antibodies and counterstained with DAPI. Crop with higher magnification of the area is indicated by the square. Pearson correlation between 1H6 and various histone modifications colocalization was calculated from the immunofluorescence pictures. The scatter plot was then divided by a K-means clustering using three groups and the Pearson coefficient is indicated on the graph. Scale bar: 10 μm. **i** Immunofluorescence pictures of HCA2 cells treated with 10 μmol/mL SB for the indicated time. Pearson correlation study of the co-expression between 1H6 and H3K9ac at the 2 h time point. Plotted in a scatter graph with the Pearson coefficient indicated on the graph. An average of 150 cells was analyzed per condition. See more quantification in Supplementary Fig. 3d, e Scale bar: 40 μm. $P \leq 0.05^*, \leq 0.01^{**}, \leq 0.001^{***}$. All values are means ± SEM. **j** IF analyses on WT and $Bmi1^{-/-}$ mouse retinal sections at P10 using the 1H6 and anti-S-Opsin antibodies. S-cone photoreceptors with the induction of G4 are showed (white arrows). Scale bars: 12 μm.

of −0.135). In contrast, H3K9ac signal intensity was increased upon *BMI1* knockdown (Fig. 1h and Supplementary Fig. 3a), and a significant positive correlation (Pearson coefficient correlation of 0.25) was observed between 1H6 and H3K9ac labeling (Fig. 1h)[45]. This suggested that the induction of G4 structures may be associated with chromatin relaxation. To test our hypothesis, we used histone deacetylase inhibitor(s) (HDACi), which lead to chromatin relaxation by preventing the deacetylation of histones[45,46]. We found that HDFs treated with sodium butyrate or trichostatin displayed rapid induction of G4 structures within 2 h, which was markedly preceded by robust elevation of H3K9ac levels (Fig. 1i and Supplementary Fig. 3b–d). Pearson correlation analyses at 2 h revealed a near-perfect correlation between H3K9ac and 1H6 labeling, suggesting that most G4 structures were induced following histone acetylation (Fig. 1i and Supplementary Fig. 3b–d).

To test if *Bmi1* knockout was also associated with the formation of G4 structures, we analyzed $Bmi1^{-/-}$ mice. The Bmi1 protein is expressed in post-mitotic retinal neurons (Supplementary Fig. 4a), including cone photoreceptors. Cones from $Bmi1^{-/-}$ mice present reduced heterochromatin compaction at post-natal (P) day 25[47,48]. In wild-type (WT) mice, we found that the baseline level of 1H6 was low in all retinal neurons (Fig. 1j and Supplementary Fig. 4a). In contrast, nuclear 1H6 immunoreactivity was high in S-opsin positive cones of $Bmi1^{-/-}$ mice (Fig. 1j, white arrows and Supplementary Fig. 4a). Notably, 1H6 induction in $Bmi1^{-/-}$ cones also correlated with increased H3K9ac level (Supplementary Fig. 4b, white arrows). Interestingly, 1H6 was also induced in cones from $Bmi1^{-/-}$ mice at P1 (Supplementary Fig. 4c), thus before the onset of retinal degeneration[48]. These results suggested that BMI1-mediated chromatin compaction prevents excessive formation of G4 structures in human and mouse cells.

**Re-establishing chromatin compaction reverses accumulation of G4 structures in AD neurons.** Considering that *BMI1* expression is reduced in AD brains and in patient-derived AD

neurons produced from induced pluripotent stem cells (iPSCs)[10], we tested whether the above findings were relevant to AD. We first evaluated the presence of G4 structures in *BMI1*-knockdown human cortical neurons, an experimental cellular model of AD[10]. We found that *BMI1*-knockdown neurons showed reduce H3K9me3 levels, which correlated with a robust accumulation of G4 structures (Fig. 2a). Using frozen brain sections from the hippocampus of three elderly controls and three AD cases, we found that G4 structures also accumulated in AD neurons in situ (Fig. 2b). Likewise, cortical neurons produced from the differentiation of iPSCs from three unrelated AD cases (AD1, AD2, and LG) presented increased H3K9ac levels that correlated with the accumulation of G4 structures, when compared to control iPSC-derived neurons from two unrelated non-demented elderly cases (Ctrl1 and Ctrl2) (Fig. 2c and Supplementary Fig. 5; see Experimental procedures for description of the iPSC lines). DNA damage accumulates in AD neurons in situ[11], and 53BP1 nuclear foci label DNA damage[49]. We found that 1H6 foci weakly colocalized with 53BP1 nuclear foci in AD neurons at day in vitro 30 (DIV30) (Supplementary Fig. 6). By performing a time-course study, we also observed the accumulation of G4 structures in AD neurons at DIV14, thus well before the appearance of 53BP1 nuclear foci at DIV30 (Fig. 2d). This suggested, at least within the timeframe analyzed herein, that G4 structures are not significantly associated with DNA damage in AD neurons.

Next, we performed high-resolution confocal microscopy and 3D reconstruction analyses on AD neurons to reveal the subcellular localization of G4 structures. G4 structures were found to be abundant around the nucleolus and at interspaced puncta located at the nuclear periphery, suggesting accumulation at the nucleolar heterochromatin and at LADs, respectively (Fig. 2e)[50,51]. Using the neuronal marker βIII-tubulin, we confirmed that G4 structures observed in AD cultures were predominant in neurons (Supplementary Fig. 5a). We further investigated if the modulation of chromatin structure could improve the observed phenotype. Tau over-expression in neurons

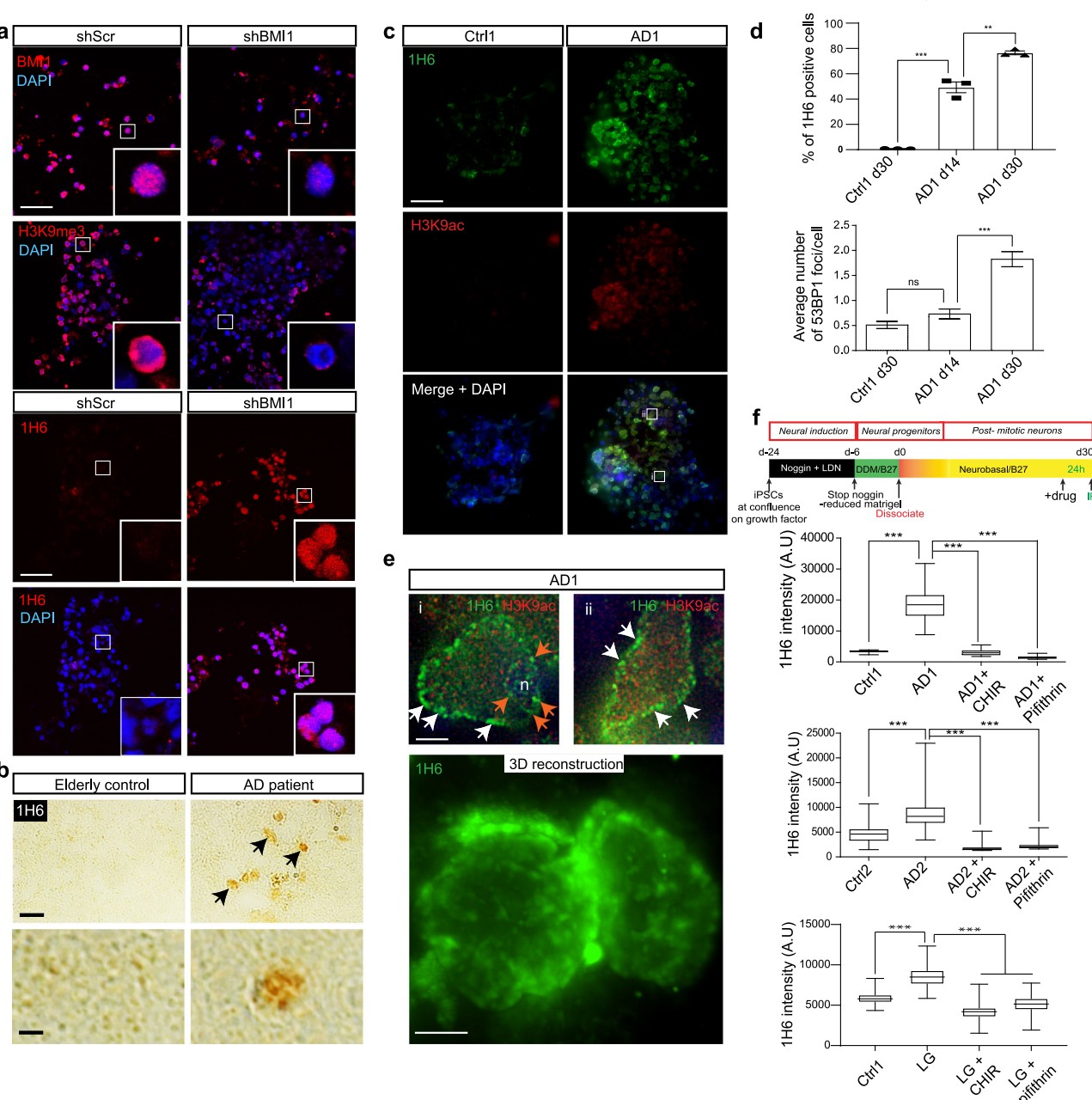

**Fig. 2 Re-establishing chromatin compaction reverses accumulation of G4 structures in AD neurons. a** IF analysis showing that BMI1 knockdown in human neurons differentiated from embryonic stem cells results in loss of heterochromatin (H3K9$^{me3}$) and induction of G4 DNA structures (1H6). Scale bar: 35 μm. **b** IHC analysis on frozen human brain sections (hippocampus) showing 1H6 immunoreactivity in AD patient's neurons. The frozen sections were from the hippocampus of end-stage disease (Braak stage V/VI) AD patients ($n = 3$) and non-demented age-matched controls ($n = 3$). The samples' sections were kindly provided by the Banner Health Institute. In the control section, there was an average of 0.75 positive nucleus/section (1H6 staining) by IHC, while in AD patients there was an average of 12.6 positive nucleus/section with a $P$ value <0.0001. **c** IF analysis showing G4 DNA structure (1H6) induction and histone H3 acetylation (H3K9$^{ac}$) in AD neurons. Scale bar: 35 μm. **d** Quantification of IF results showing G4 DNA induction in AD neurons that occurred before 53BP1 accumulation. Statistical differences were analyzed using an unpaired $t$-test with two tails, and the result is shown on the graph as mean ± SEM. $P \leq 0.05*$, $\leq 0.01**$, $\leq 0.001***$. For the percentage of 1H6 neurons a total of three fields were quantified in each condition, each field contained 300 cells. For the number of foci per cell a total of 300 cells were analyzed. **e** High-magnification IF analysis showing G4 DNA structures at the nuclear membrane (white arrows) and nucleolar (n) periphery (orange arrows) in AD neurons. This phenotype can be also visualized using 3D reconstruction. Scale bar: 10 μm. **f** Schematic of the method used to produce iPSC-derived neurons (top image). Control (Ctrl) and AD iPSC-derived cortical neurons were treated for 24 h with an inhibitor of GSK3β (CHIR99021) or an inhibitor for p53 (pifithrin). They were then labeled with the 1H6 and H3K9$^{ac}$ antibodies and counterstained with DAPI prior to immunofluorescence analysis. Mean 1H6 or H3K9$^{ac}$ fluorescence intensity/cell was quantified from Supplementary Fig. 5b–d and plotted in a whisker and box plot. Statistical differences were analyzed using an unpaired $t$-test with two tails. $P \leq 0.001***$. All values are mean ± SEM (see Supplementary Fig. 5b–d). The box plots show the minima and the maxima values delimited by the whisper, the box delimits the 25th percentile and 75th percentile, and the line inside the box define the mean. A total of 200 cells were quantified in each condition.

is sufficient to induce heterochromatin relaxation, and GSK3β is the main kinase that phosphorylates Tau in AD[12,52,53]. Likewise, p53 accumulates in AD neurons, and p53 can initiate heterochromatin relaxation by inhibiting the expression of the histone Lys9 tri-methyltransferase *SUV39H1*[10,54]. We thus treated control and AD neurons for 24 h with CHIR99021 (a GSK3β inhibitor) or pifithrin-α (a p53 inhibitor) (Fig. 2f). We found that both treatments significantly reduced H3K9 hyper-acetylation and the accumulation of G4 structures in neurons from the three AD cases (Fig. 2f and Supplementary Fig. 5b–e), revealing that re-establishment of chromatin compaction is sufficient to rescue the G4 phenotype.

**1H6 peaks present high similarities with canonical and non-canonical G4 sequences.** In order to localize G4 structures on the human genome, we performed ChIP-seq on control and AD neurons at DIV30 using the 1H6 antibody[10]. We identified 1389 peaks in AD neurons and 1165 peaks in control neurons. Enrichment of 1H6 within the peaks was significantly higher in AD neurons (Fig. 3a and Supplementary Fig. 7a). Peaks in AD neurons were also significantly larger than in controls, covering ~0.6% of the human genome, compared to less than 0.2% (Fig. 3b). While most of the peaks in control neurons were represented in AD samples (Supplementary Fig. 7a), cluster 1 from the AD peaks showed no enrichment in control neurons, revealing a large subgroup of AD-specific peaks (Fig. 3a). Venn diagram distribution revealed that out of the 1389 AD peaks, 737 peaks were AD-specific, and 558 were shared with control neurons. Out of the 1165 control peaks, 504 peaks were control-specific, and 609 were shared with AD neurons (Fig. 3c). The discrepancy in the number of peaks shared by both groups was explained by the occasional presence of two control peaks within a very large and unique AD peak.

We observed by immunofluorescence that most G4 structures induced in AD neurons localized at LADs. Bioinformatic analysis of ChIP-seq data further revealed that ~40% of 1H6 peaks colocalized with constitutive LAD (cLADs) and ~25% with facultative LAD (fLADs) (Fig. 3e)[55]. AD peaks were also found to colocalized more with cLADs than control peaks. Hence, ~75% of AD peaks indeed colocalized with cLADs or fLADs. The observed percentage could not be explained by the coverage of these domains on the genome since cLADs and fLADs represent ~50% of the human genome (Fig. 3f).

We next compared all 1H6 peaks with the canonical G4 motif. The most stringent predicted canonical G4 sequence contains four interspaced repeats with at least three guanines/repeat, the guanine repeats being separated by a loop of 1–12 nucleotides[56]. This revealed that 61% of 1H6 peaks in AD neurons and 53% of 1H6 peaks in control neurons colocalized with a predicted canonical G4 sequence (Supplementary Fig. 7b). Importantly, these values presented a Z score nine times higher than a random distribution of all peaks on the human genome when performing a test with 1000 random permutations (Supplementary Fig. 7c). Annotation of all 1H6 peaks revealed that ~38% were located within gene bodies, ~20% at enhancers, and ~5% at CpG islands (Fig. 3d). We also analyzed all peaks using an unbiased motif discovery algorithm. We found that the most statistically significant motifs discovered by MEME Suite contained interspaced G repeats predicted to form non-canonical G4 structures (Fig. 3g, h and Supplementary Fig. 7e)[57,58]. When probed with a QGRS finder (quadruplex forming G-rich sequences)[59], these motifs presented a *G* score of 34[60]. When analyzing with MEME the peaks that did not colocalize with a canonical G4 sequence (~45% of all peaks), we found that the most statistically significant motifs also contained interspaced G repeats predicted

to form non-canonical G4 structures (Supplementary Fig. 7f). These results indicate that most epitopes recognized by the 1H6 antibody in the ChIP-seq experiment corresponded to structured DNA.

**1H6 peaks correspond to evolutionary conserved L1 sequences.** Using bioinformatic analyses, we found that the best clustering approach to segregate 1H6 peaks was obtained when testing for the enrichment of repetitive elements. Indeed, upon annotation of all peaks for the presence of repetitive elements, we found that Long Interspersed Elements (LINEs) were present in 95% of peaks from control neurons and in 98% of peaks from AD neurons (Fig. 4a). Furthermore, 75% of the peaks also contained at least one Short Interspersed Element (SINE) (Fig. 4a). Notably, the L1 family was the most enriched for LINEs, and the ALU family was the most enriched for SINEs (Fig. 4b, c). The enrichment for LINEs and SINEs in all 1H6 peaks was much higher than the theoretical percentage obtained when searching for LINEs and SINEs in all predicted G4 sequences of the human genome (Fig. 4a–c, showed with blue histograms). These differences in enrichment could also not be accounted for by a difference in genome coverage between the mean of control and AD 1H6 peaks (~0.3%) and all predicted G4 sequences (~0.3%) (Fig. 3b). Furthermore, we performed a permutation test with 1000 permutations of the peaks and compared them to the various repetitive families and found a significant difference between the random set and the 1H6 peaks giving with a *P* value associated with this difference (Fig. 4a–c and Supplementary Fig. 8, ***P < 0.001). Among the L1 family, the LIPA3, LIPA2, LIM5, and LIMC4 sequences were the most represented (Fig. 4d). Among the ALU family, the AluSx, AluY, AluSz, and AluSxz sequences were the most represented (Supplementary Fig. 9a). Remarkably, when searching for predicted G4 motifs in the consensus sequences for these repetitions, we found that each contained at least three canonical G4 sequences (Fig. 4e and Supplementary Fig. 9b).

Considering the above findings, we investigated the presence of LINEs or SINEs in intragenic 1H6 peaks. Notably, a unique peak was identified in both control and AD neurons between exons 13 and 14 of the *Amyloid beta-precursor protein* (*APP*) gene. In AD neurons, the peak was much larger than in control neurons, spanning two extra-canonical G4 sequences (Fig. 4f). We also noticed the presence of two regions with very high reads density within this unique peak (Fig. 4f, boxed peaks in red). Further analysis revealed that these regions contained four non-canonical G4 sequences presenting a relatively high G4 score (Fig. 4f). Strikingly, we found a near-perfect match between the presence of a large intragenic 1H6 peak and the presence of a unique and evolutionary conserved L1 sequence at the *APP* locus and all other tested loci (Fig. 4f and Supplementary Fig. 9c). While SINEs were frequently present within or close to 1H6 peaks, they were not as predominant as L1 sequences (Fig. 4f and Supplementary Fig. 9c). Noticeably, the presence of "bystander" SINEs apparently accounted for the enlarged 1H6 peak found in AD neurons at the *APP* locus (Fig. 4f). These findings suggested that evolutionary conserved L1 sequences represent the source of almost all G4 structures detected in healthy and AD neurons.

Evolutionary conserved L1 sequences are the only LINEs with intact internal promoters, allowing them to be transcribed by RNAPII. To test if the formation of G4 structures was dependent on transcription, we treated control and AD neurons with the RNAPII inhibitor 5,6-dichlorobenzimidazole 1-β-D-ribofuranoside (DRB) for 8 h prior to immunofluorescence analysis (Fig. 4g and Supplementary Fig. 10a, b). We found that inhibition of RNAPII mildly impacted G4 structures in control neurons, but dramatically

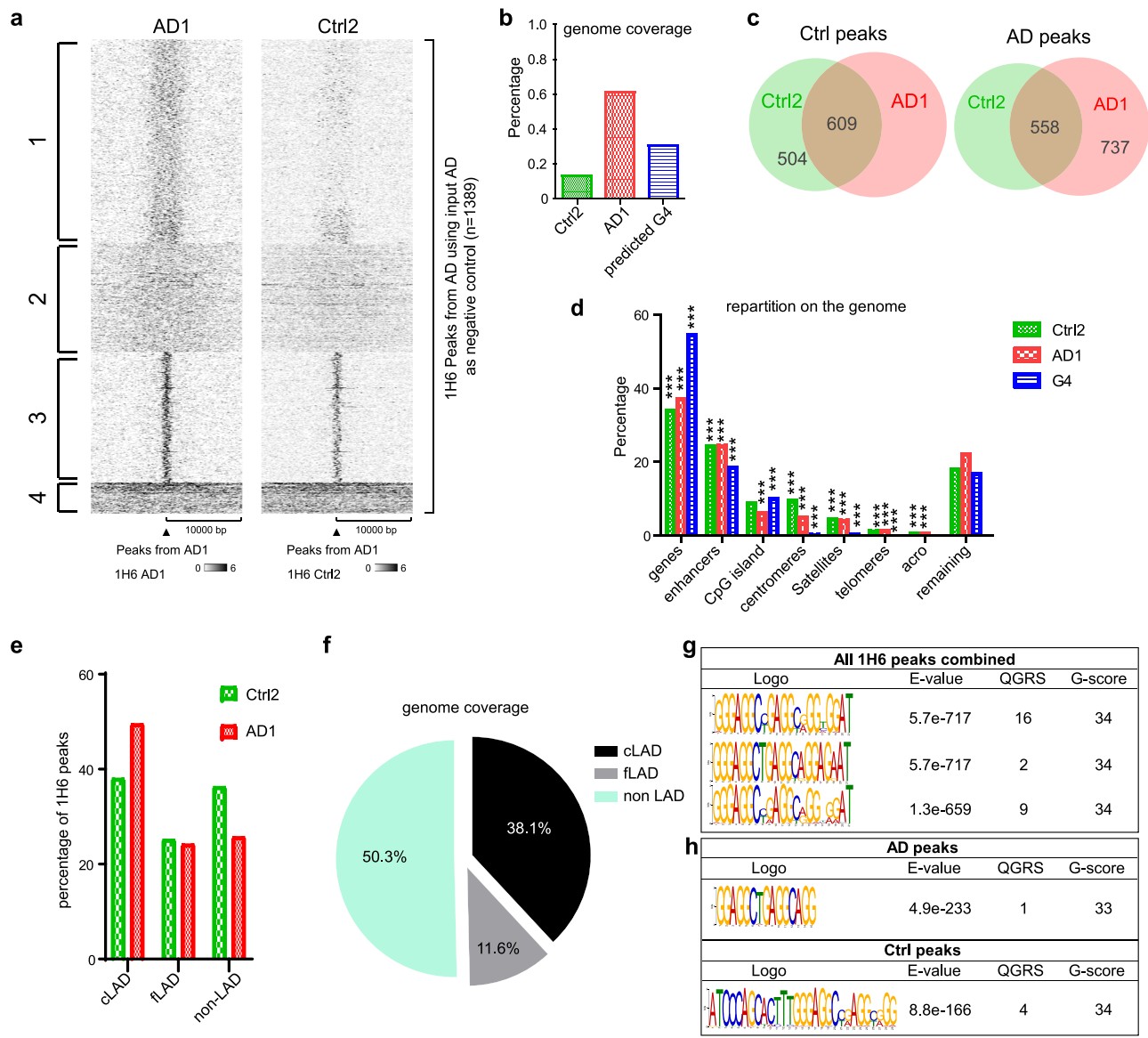

**Fig. 3 1H6 peaks present high similarities with canonical and non-canonical G4 sequences. a** Heatmap of 1H6 ChIP-seq enrichment from AD1 neurons or control neurons (Ctrl2) centered on AD1 peaks with a ±10 Kbp. K-means clustering highlighted on the right was done using the AD1 enrichment. **b** Histogram showing the percentage coverage of 1H6 peaks and predicted G4 sequences on the entire human genome. **c** Venn diagram shows the number of AD1 peaks that are specific and those that are shared with a Ctrl2 peak and a diagram that shows the Ctrl2 peaks that are specific and those that are shared with an AD1 peak. **d** Repartition of 1H6 peaks and predicted G4 sequences on the genome. A permutation test was performed for each combination in order to calculate the significance. A total of 1000 permuted sets of probes were randomly generated and annotated for the different parts of the genome. **e** Graph showing the percentage of colocalization between 1H6 peaks and LAD or non-LAD. **f** A pie chart showing the repartition of the genome between LAD and non-LAD. **g** MEME analysis on all the ChIP-seq combined on the neurons showing the top three motifs *E* value associated with them, the number of QGRS (quadruplex forming G-rich sequences), and the highest *G* score associated with these sequences. **h** MEME analysis on the 1H6 peaks of Ctrl2 or AD1 neurons showing motifs that can form a G4 structure with the *E* value associated with them, the number of QGRS, and the highest *G* score associated with these sequences.

reversed the accumulation of G4 structures in AD neurons from the 3 AD cases (Fig. 4g and Supplementary Fig. 10a, b). Accordingly, the 1H6 signal largely colocalized with that of RNAPII in AD neurons (Pearson correlation analysis: 0.63) (Fig. 4h). To confirm this, we treated control neurons with HDACi or HDACi + DRB. While HDACi treatment resulted in the induction of G4 structures and H3K9 acetylation, the addition of DRB prevented the induction of G4 structures independently of H3K9 acetylation (Fig. 4i and Supplementary Fig. 10c). These results thus distinguished RNAPII-mediated G4 induction from chromatin relaxation.

**G4 structures can perturb splicing and gene expression in AD neurons**. To evaluate the biological significance of our findings, we performed a Gene Ontology (GO) analysis of genes associated with 1H6 peaks. Among the 139 control-specific peaks associated with a gene (i.e., lost in AD), there was no pathway enrichment. Among the 259 genes with peaks common to control and AD neurons (i.e., common peaks), there was an overrepresentation of genes involved in de-ubiquitination (Fig. 5a). Among the 269 AD-specific peaks associated with a gene (i.e., gained in AD), there was a general overrepresentation of genes involved in cell–cell adhesion, axonal projection, neurogenesis, and

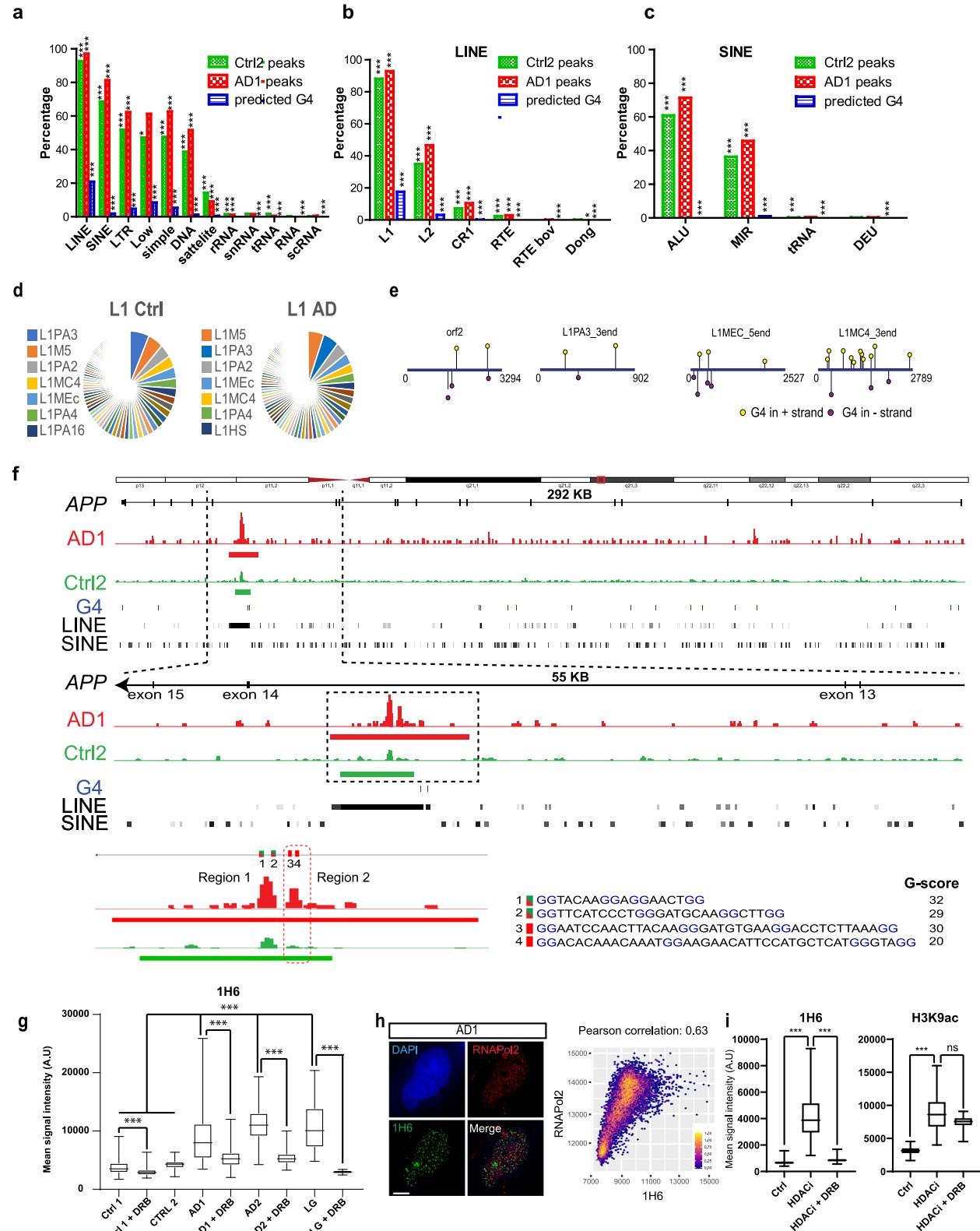

synaptogenesis (Fig. 5a). Notably, comparative RNA-seq analysis of Ctrl1 and Ctrl2 neurons with AD1 and AD2 neurons revealed that a subset of genes with intergenic or intragenic peaks presented modified splicing events (Fig. 5b, c and Supplementary Fig. 11a, b). In the first scenario (lost in AD), a single intergenic peak between *SMN1* and *NAIP* was lost in AD neurons. This resulted in cryptic slicing and/or transcription from *SMN1* over

the adjacent *NAIP* locus (Fig. 5b), possibly explaining the presence of aberrant and larger SMN1 protein isoforms in AD neurons (Fig. 5d and Supplementary Fig. 12). In the second and third scenarios (common peaks and gained in AD), the presence of a single intragenic peak was found to be closely associated with alternative splicing of minor isoforms and/or exon exclusion events (Fig. 5b), possibly explaining perturbations in APP and

**Fig. 4 1H6 peaks correspond to evolutionary conserved L1 sequences. a** Histogram showing the distribution of 1H6 peaks and predicted G4 sequences within families of repetitive elements present in the human genome. One peak can contain a combination of various repeat elements. A permutation test was done for each combination in order to calculate the significance. A total of 1000 permuted sets of probes were randomly generated and annotated for the repeat elements. ***$P < 0.001$, Student's unpaired $t$-test with two tails. **b** Histogram showing the distribution of 1H6 peaks and predicted G4 sequences within members of the LINE family. One peak can contain a combination of various repeat elements. A permutation test was done for each combination in order to calculate the significance. A total of 1000 permuted sets of probes were randomly generated and annotated for the repeat elements. ***$P < 0.001$, Student's unpaired $t$-test with two tails. **c** Histogram showing the distribution of 1H6 peaks and predicted G4 sequences within members of the SINE family. One peak can contain a combination of various repeat elements. A permutation test was done for each combination in order to calculate the significance. A total of 1000 permuted sets of probes were randomly generated and annotated for the repeat elements. ***$P < 0.001$, Student's unpaired $t$-test with two tails. **d** Pie chart showing the repartition of repeats from L1 sequences—the most represented LINE family in AD1 and Ctrl2 peaks. **e** Graphical representation of the position of all predicted G4 sequences prevalent in the L1 family. The consensus sequence for each sub-class of L1 was used. **f** Physical map showing enrichment of 1H6 peaks (red: AD neurons; green: Ctrl neurons) within a unique region of the *APP* locus and spanning 6–7 kb of genomic DNA. Predicted canonical G4 sequences are shown in blue. LINEs and SINEs present at the *APP* locus are represented in shades of grays. The shade of grays within LINEs or SINEs reflects their degree of conservation, with black being the most conserved. Lighter shades indicate the presence of base mismatch, base deletion, and base insertion. A zoom of the peak is represented and showing two regions of high reads and containing several G4 motifs. We used the Quadparser algorithm to measure the corresponding G score of each motif. **g** Immunofluorescence analysis of iPSC-derived neurons treated or not with DRB for 8 h, as presented in (**h**). Neurons were fixed with paraformaldehyde and immunolabeled with 1H6 before counterstaining with DAPI. The associated graphs show the quantification of 1H6 fluorescence intensity/cell from Supplementary Fig. 9a, b. All values are means ± SEM. ***$P < 0.001$, Student's unpaired $t$-test. with two tails (see Supplementary Fig. 9a, b). The box plots show the minima and the maxima values delimited by the whisper, the box delimites the 25th percentile and 75th percentile, and the line inside the box define the mean. A total of 150 cells were quantified in each condition. **h** Formaldehyde fixed AD neurons were immunolabeled with RNAPII and 1H6, and counterstained with DAPI. A zoom of a single cell is shown. Pearson correlation analysis was conducted on the AD samples. This revealed a strong correlation (0.63) between RNAPII and 1H6. Scale bar: 10 µm. **i** Mean signal intensity of 1H6 and H3K9$^{ac}$ in nucleus from neurons treated with HDACi, or HDACi + DRB. Values were measured from Supplementary Fig. 9c and plotted in a box and whisker graph. Statistical differences were analyzed using unpaired $t$-test with two tails. Scale bar: 35 µm. $P \le 0.05*$, $\le 0.01**$, $\le 0.001***$ (see Supplementary Fig. 9c). The box plots show the minima and the maxima values delimited by the whisper, the box delimites the 25th percentile and 75th percentile, and the line inside the box define the mean. A total of 150 cells were quantified in each condition.

DNAH6 protein isoforms observed in AD neurons (Fig. 5d and Supplementary Fig. 12). Efficient exon splicing can be associated with increased gene expression, while exon exclusion results in the opposite trend, a process called exon-mediated activation of transcription starts[61]. In addition, G4 structures located in intragenic regions have been shown to represent an obstacle to transcriptional elongation, resulting in reduced gene expression[62].

To test if our results matched any of these models, we superimposed all genes having a peak (lost in AD, common and gained in AD) over a Volcano plot distribution of all genes differentially expressed between control and AD neurons. While genes were generally upregulated in AD neurons (2006 upregulated and 1924 downregulated in AD1; 9584 upregulated and 5851 downregulated in AD2), no significant correlation was observed between the presence of a G4 structure and differential gene expression (Supplementary Figs. 13a and 14a). However, we observed a trend toward gene downregulation in genes having a common peak or a peak only present in AD neurons (Supplementary Figs. 13b and 14b). When we analyzed genes with an intragenic G4 structure that was also associated with an alternative splicing event, we found that these were significantly downregulated in AD neurons (Fig. 5e and Supplementary Figs. 11c, 13c, and 14c). These results suggested that intergenic G4 structures may sometime work as "gene insulator" elements in normal conditions and that excessive formation of intragenic G4 structures can perturb alternative splicing and gene expression in AD neurons.

## Discussion

We report here that *BMI1* inactivation in human cells or mouse photoreceptors resulted in heterochromatin relaxation and induction of G4 structures. A similar phenotype was observed in AD neurons having reduced *BMI1* expression. Loss of chromatin compaction in general, as shown using HDACi, also resulted in the induction of G4 structures. Conversely, the formation of G4 structures in AD neurons was reversed by re-establishing chromatin compaction. ChIP-seq analysis using 1H6 further revealed that most significant consensus motifs contained canonical and non-canonical G4 sequences and that 1H6 peaks corresponded to evolutionary conserved L1 sequences. Hence, inhibition of RNAPII allowed us to distinguish transcription-mediated induction of G4 structures from chromatin relaxation. 1H6 enrichment was predominant in AD neurons, and most AD-specific peaks were linked to genes involved in neurogenesis and synaptogenesis. At last, some intragenic peaks enriched in AD neurons were associated with alternative splicing events, exon exclusion, and reduced gene expression.

Herein, we have preferentially used the 1H6 antibody, which was shown to recognize several, but not all, G4 structures in vitro[35]. We have further validated the specificity of this antibody using several approaches and by showing: (1) specific labeling after pyridostatin treatment, but not after gamma irradiation; (2) colocalization with the Werner and XPB helicases; (3) equivalent results when using the BG4 antibody; (4) specific labeling only under native chromatin conditions; and (5) detection of canonical and non-canonical G4 sequences in ChIP-seq consensus motifs. Previously, high-throughput sequencing of indirect DNA immunoprecipitation experiments with the BG4 antibody suggested that G4 DNA-forming sequences were abundant in gene bodies[63,64]. ChIP-seq analyses using BG4 further confirmed that, under physiological conditions, G4 structures were enriched at the promoter of some actively transcribed genes, such as C-MYC[34]. These observations are consistent with our findings in control neurons, where several G4 peaks were detected at intragenic and enhancer regions.

In AD neurons, heterochromatin compaction was reduced, and we found that G4 structures preferentially accumulated at LADs and peri-nucleolar heterochromatin. Likewise, heterochromatin compaction was reduced and G4 structures were induced in *BMI1*-knockdown neurons. Notably, it was found that most LINEs are localized at LADs[65], and we indeed found that about all 1H6 peaks corresponded to L1 sequences. In this context, it is notable that BMI1 co-purifies with LMNA and LMNB2, alongside with architectural heterochromatin proteins[9], suggesting that

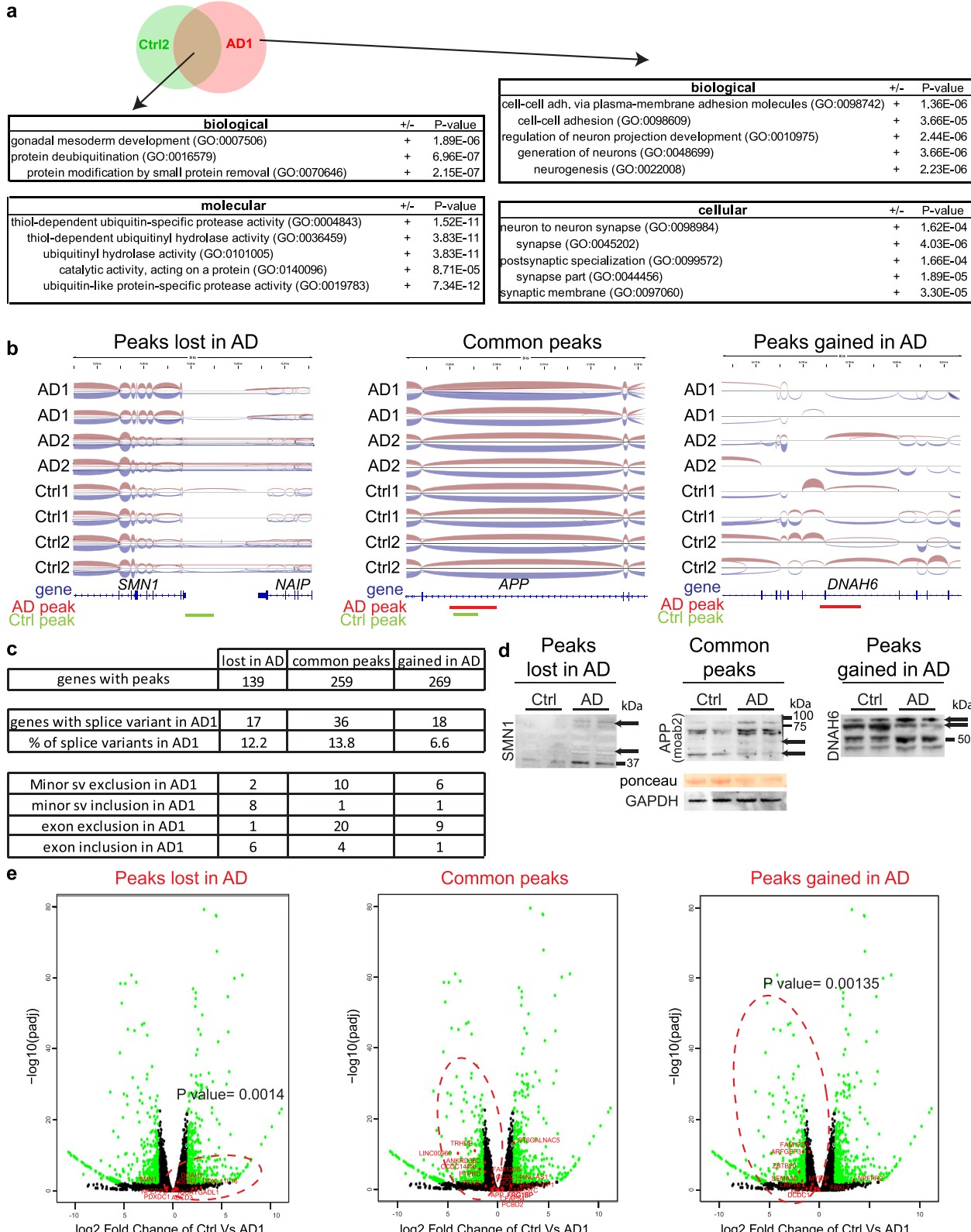

BMI1 may, directly or indirectly, be important for the stabilization of LADs. L1 sequences represent nearly 17% of the human genome[66]. While these sequences can be autonomously transcribed by RNAPII from a 5' UTR promoter, it is estimated that out of the 500,000 L1 sequences present in the human genome, less than 5000 have an intact internal promoter[67]. Notably, it was found that the 3' UTR of L1 sequences can form G4 structures

in vitro and that this feature is conserved among mammals, suggesting a possible functional role for the propagation of retrotransposons[68,69]. Strikingly, we have found that only evolutionary conserved L1 sequences, i.e., predicted to be transcriptionally active, were associated with 1H6 peaks. This is consistent with our observations that the accumulation of G4 structures required functional RNAPII activity. We also showed that upon RNAPII

**Fig. 5 1H6 peaks are enriched at key regulatory elements of neuronal genes. a** Venn diagram and Gene Ontology analysis of genes associated with peaks common to both Ctrl2 and AD1 neurons or specific to AD neurons (AD1). The most significant pathways are indicated on the diagram along with the respective *P* value calculated by the GO algorithm. **b** Sashimi plot of three different loci containing a 1H6 peak and associated with a differential splicing event. Exons are represented as blue boxes. The red lines represent a peak found in AD neurons; the green lines represent a peak found in control (Ctrl) neurons green. Loss of the intragenic peak in AD neurons between the *SMN1* and *NAIP* genes is associated with the formation of cryptic fusion transcripts. The common peak found at the *APP* locus (but increased in AD) is associated with loss of some minor isoforms in AD neurons. The unique peak gained in AD neurons at the *DNAH6* locus is associated with loss of minor isoforms and a large splicing gap in AD neurons. **c** Table showing the number of genes containing a 1H6 peak, as well as the number of genes that entailed a differential splicing event associated with a 1H6 peak when comparing Ctrl1 and Ctrl2 neurons with AD1 neurons. The splicing events were subdivided into loss of a minor isoform or formation of a gap junction. **d** Western blot analyses of Ctrl2 and AD2 neurons. GAPDH and ponceau were used as normalizers. Black arrows indicate differentially expressed protein variants. **e** Volcano plot showing differential gene expression between control (Ctrl1 and Ctrl2 combined) and AD1 neurons using RNA-seq. Superimposed in red are genes containing a G4 peak and associated with an abnormal splicing event. Note that genes in the third volcano plot (i.e., gained in AD) are significantly downregulated, and genes in the first volcano plot (i.e., lost in AD) are significantly upregulated in AD1. For each gene distribution (red dotted ovals), a *P* value was calculated using binomial distribution, knowing that 51% of the genes were upregulated.

inhibition, chromatin relaxation could be uncoupled from the accumulation of G4 structures. This supports a model where chromatin-mediated inhibition of L1 sequences transcription is the primary mechanism regulating the formation of G4 structures in post-mitotic neurons.

A recent study revealed that many genes in AD brains displayed alternative splicing events[70], but the underlying mechanism remained unknown. Nevertheless, part of these splicing anomalies was mimicked in neurons over-expressing Tau[70], thus possibly linking Tau-mediated chromatin relaxation to specific alternative splicing events. Here, we found that some intragenic 1H6 peaks enriched in AD neurons were associated with alternative splicing events and reduced gene expression, thus providing a plausible mechanism for a fraction of genes with perturbed splicing in AD. The overrepresentation of 1H6 peaks in genes involved in neurogenesis, axonal guidance, and synaptogenesis in AD samples is also intriguing considering the reported inability of AD neurons to guide their projections and form functional synapses[71–78]. It is also intriguing considering that most genes with an exon-mediated activation of transcription starts structure are also enriched for brain development, neurogenesis, and synaptogenesis[61]. Further work is required to better understand the complex interconnections between G4 structures, splicing, and neuronal gene expression.

In conclusion, while specialized DNA helicases have evolved to recognize and resolve G4 structures to promote transcription, replication, and repair, chromatin-mediated transcriptional repression of L1 sequences represents a critical mechanism to prevent excessive formation of G4 structures in human neurons. Considering that heterochromatin relaxation in neurons is also observed in other neurodegenerative diseases, such as amyotrophic lateral sclerosis, frontotemporal dementia, and tauopathies[12,19,79–83], these findings have potentially broader implications. Our findings thus expose, for the first time, the nature of DNA sequences capable of forming G4 structures in human neurons, the mechanisms underlying their activation, and the biological impact of their deregulation.

## Methods

**Human samples and animals.** Human pluripotent stem cells were used in accordance with the Canadian Institute Health Research (CIHR) guidelines and approved by the "Comité de Surveillance de la Recherche sur les Cellules Souches" of the CIHR and Maisonneuve-Rosemont Hospital Ethic Committee. Human brain tissues were kindly provided by the Banner Institute. C57Bl/6 *Bmi1*[−/−] (The Netherlands Cancer Institute, Amsterdam) and WT (Charles River, St-Constant, Canada) mice were used in accordance with the Animal Care Committee of the Maisonneuve-Rosemont Hospital Research Centre (Approval ID #2014-03, #2012-09). The mice that were used were equally distributed between male and females and they were sacrifices at P1 and P10 for the respective experiments. These mice were kept at the HMR animal facility, with ambient temperature around 22 °C, controlled humidity with a ventilation system and a 12-h dark/light cycle.

**Cell cultures.** Normal HDFs were purchased from the Coriell Institute. HCA2 cells were kindly provided by the laboratory of Dr Francis Rodier. HDFs were cultured with DMEM/F12 media (Invitrogen) supplemented with 10% FBS (Invitrogen) and non-essential amino acids (Invitrogen). For the HDACi experiments, cells were treated with 5 ng/ml of Trichostatin A (Sigma, T1952-200UL) or 10 mM of Sodium Butyrate (Sigma, 303410-5G), and 5 μM of pyridostatin (Sigma, SML0678-5MG). For the replication and transcription arrest we used respectively: 1 μg/ml of Aphidicolin from Nigrospora sphaerica (Sigma, A0781-1MG) and 40 μM of DRB (Sigma, D1916-10MG). A concentration of 0.2 μg/ml of aphidicolin or of 0.2 mM of Hydroxyurea (Sigma, H8627) was used to induce replication stress. iPSC and neuronal cultures were conducted according to the methods described in[10].

**Differentiation of pluripotent stem cells into cortical neurons.** Differentiation of iPSCs into cortical neurons was performed accordingly to Flamier et al.[10].

Ctrl1: iPSC-derived fibroblasts from Coriell Institute #AG04152, from a healthy 82-year-old male patient with no family history of AD.

Ctrl2: iPSC-derived fibroblasts from Coriell Institute #AG09602 from a healthy 92-year-old female patient with no family history of AD.

AD1: iPSC-derived fibroblasts from Coriell Institute #AG08243 from a 72-year-old male patient diagnosed with sporadic AD with no family history of AD.

AD2: iPSC-derived fibroblasts from Coriell Institute #AG08259 from a 90-year-old male patient diagnosed with sporadic AD at the age of 87 with no family history of AD.

LG[84]: iPSC-derived fibroblasts from Coriell Institute #GM24666 from an 83-year-old male patient diagnosed with sporadic AD at the age of 78 with no family history of AD with a 3/3 *APOE* genotype.

iPSCs were dissociated using Accutase (Innovative Cell Technology #AT-104) and platted on growth factor reduced Matrigel (Corning #356231) in PeproGrow hES cell media (PeproTech #BM-hESC) supplemented with ROCK inhibitor (Y-27632;10 μM, Cayman Chemical #10005583). Upon 70% of confluency, the medium was changed to DDM supplemented with B27 (1X final), Noggin (10 ng/ml, PeproTech #120-10C) and LDN193189 (0.5 μM; Sigma #SML0559). The medium was changed every day. After 16 days of differentiation as neural progenitors, the medium was changed to DDM/B27 and replenished every day. At day 24 as neural progenitors (corresponding to DIV0 of neural differentiation), half of the medium was changed for Neurobasal A media supplemented with B27 (1X final) and changed again every 3 days. In Fig. 2a, neurons produced from the differentiation of the HUES9 cell line were infected at DIV0 with a lentivirus expressing an shRNA targeting BMI1 or a scramble sequence, as described in Flamier et al.[10].

**Immunofluorescence.** Eyes were extracted and fixed by immersion overnight at 4 °C in 4% PFA/3% sucrose in PBS, pH 7.4. Samples were washed three times in PBS, cryoprotected in PBS/30% sucrose, and frozen in CRYOMATRIX embedding medium (Thermo Shandon, Pittsburgh, PA) or in Tissue-Tek® optimum cutting temperature compound (Sakura Finetek, USA). Then, 5- to 12-μm thick sections were mounted on Super-Frost glass slides (Fisher Scientific) and processed for immunofluorescence. For immunofluorescence labeling, sections were incubated overnight with primary antibody solutions at 4 °C in a humidified chamber. After three washes in PBS, sections were incubated with secondary antibodies for 1 h at room temperature. Slides were mounted on coverslips in DAPI-containing mounting medium (Vector Laboratories CA, H-1200). Confocal microscopy analyses were performed using 60× objectives with an IX81 confocal microscope (Olympus, Richmond Hill, Canada), and images were obtained with Fluoview software version 3.1 (FV10-ASW V. 3.1, Olympus). The cultured cells were fixed for 15 min with 4% PFA, washed three times and then permeabilized for 10 min with 0.25 Triton (Sigma, X100-500ML), cells were then blocked in PBS/2% BSA (Sigma, A7906-100G) for an hour and incubated overnight with the primary antibody. Primary antibodies used in this study are: FITC mouse anti-TRA-1-60

(BD Pharmingen, 560380), rabbit anti-SOX2 (ab97959), goat anti-NANOG (R&D systems, af1997), goat anti-S-Opsin (1:250, Santa Cruz, sc-14363), rabbit anti-H3K9me3 (1:500, Abcam, ab8898), rabbit anti-H3K9ac (1:500, Cell Signaling, 9671S), rabbit anti-WRN (1:100, Santa Cruz, sc-5629), rabbit anti-TFIIH p80 (1:200, Santa Cruz, sc-20696) targeted against XPD, rabbit anti-TFIIH p89 (1:200, Santa Cruz, sc-293) targeted against XPB, rabbit anti-53BP1 (1:100, Novus, NB100-304), rabbit anti-H2Aub (1:200, Cell Signaling, 8240S), rabbit anti-Ki67 (1:1000, Abcam, ab15580), and mouse BG4 (1:333, Absolute antibody, Ab00174-1.1), and 1H6 antibodies recognizing G-quadruplexes. We obtained the 1H6 antibody from The European Research Institute for the Biology of Ageing. After the primary antibodies, slides were washed three times using PBS and incubated with the secondary antibodies for 1 h. The Secondary antibodies are: donkey AlexaFluor488-conjugated anti-mouse (1:1000, Life Technologies, A21202), donkey AlexaFluor488-conjugated anti-rabbit (1:1000, Life Technologies, A21206), donkey AlexaFluor647-conjugated anti-mouse (1:1000, Life Technologies, A31571), goat AlexaFluor Texas red-conjugated anti-rabbit (1:1000, Life Technologies, T2767). Slides were then washed three times with PBS and mounted with coverslips in DAPI-containing mounting medium (Vector Laboratories CA, H-1200).

**Quantifications and statistical analysis and reproducibility**. For the colocalization study, random lines were drawn on individual cells using FIJI. From these lines, we plotted the intensity profile of each marker accordingly. The data collected were plotted on horizontal graphs with each marker as a separate line for the visualization of the peaks (example Fig. 1h). The data were also plotted in a scatter graph, using GraphPad Prism 5, to visualize the correlation between these two markers. On these sets of pairs, a Pearson correlation was calculated to quantify the correlation. For the co-expression study intensity of the signal for different markers was measured, using a mask on DAPI to identify the nucleus, and then plotted in a scatter plot using GraphPad Prism 5, to visualize the correlation between these two markers. On these sets of pairs, a Pearson correlation was calculated to quantify the correlation. For the expression study, we quantified the mean intensity of each marker in the nucleus area using the DAPI signal to identify that area. Values were plotted with a box and whisker graph. The K-means clustering was done searching for two or three groups with 20 iterations for each run. Statistical differences were analyzed using Student's t-test for unpaired samples with $P \leq 0.05^*$, $\leq 0.01^{**}$, $\leq 0.001^{***}$. All the IF experiments (Fig. 1d–j, 2a, c–f, and 4g–i and Supplementary Figs. 1–6 and 10) were repeated a minimum of three independent times with consistent results.

**Western blot**. Cell extracts were homogenized in the Complete Mini Protease inhibitor cocktail solution (Roche Diagnostics), followed by sonication. Protein material was quantified using the Bradford reagent. Proteins were resolved in 1× Laemlli reducing buffer by SDS-PAGE electrophoresis and transferred to a Nitrocellulose blotting membrane (Bio-Rad). Subsequently, membranes were blocked for 1 h in 5% non-fat milk-1X TBS solution and incubated overnight with primary antibodies. The antibodies used in this study are: mouse anti SMN (1:1000, Santa Cruz, SC-32313), rabbit anti-MOAB (1:1000, Novus, NBP2-13075), mouse anti-B4 (1:1000, Santa Cruz, SC-28365), rabbit anti-ANOA4 (1:1000, Invitrogen, PA5-62785), and rabbit anti-DNAO6 (1:1000, Invitrogen, PA5-57636). Membranes were then washed three times in 1X TBS; 0.05% Tween solution and incubated for 1 h with corresponding horseradish peroxidase-conjugated secondary antibodies. Membranes were developed using the Immobilon Western (Millipore).

**Public ChIP-seq analysis and prediction of G-quadruplexes**. ChIP-seq data sets were obtained through the GO platform using accession numbers GSE22162 (ATRX and H3K9me3), GSE44849 (XPB and XPD), GSE76688 (BG4), and GSE38273 (BMI1). BMI1 significant peaks were extracted from Meng et al.[85]. The Model-based Analysis for ChIP-Seq was used to extract significant peaks with a P value cutoff ≤0.05. Peak coordinates were mapped onto hg19 genome reference using SeqMonk v0.34.0 software (Babraham Bioinformatics). Putative G-quadruplexes were predicted using Quadparser algorithm V2 running under python v2.7.11 with indicated parameters for the number of guanines in each stack (G-groups), the number of base pairs between G-groups (loop size), and the number of time the loop and a stack was repeated after the initial stack (Repeats-1)[33]. For example, to perform a Quadparser on chromosome 1 searching for a stack of five Gs and a loop size of 1–7 with four repeats the scrip and the parameters will be: quadparser.py -f chr1.fa -r ([gG]{5,}\w{1,7}){3,}[gG]{5,}. G-quadruplexes coordinates for each set of parameters were then mapped onto hg19 genome reference using SeqMonk software. Annotation of ChIP-seq peaks with G-quadruplexes was determined by extending them 50 base pairs on each side and counting the number of overlapping predicted G-quadruplexes. SeqMINER was used for H3K9me3 ChIP-seq enrichment heatmap and K-means clustering using default parameters.

**ChIP-seq experiment and data analysis**. The ChIP-seq experiment was done using the Diagenode kit: "iDeal ChIP-seq kit for Histones" reference number: C01010051. We followed the manufacturer protocol with a minor change: after cross-linking, we added pyridostatin to buffer B in order to stabilize already present

G4 structures. Immunoprecipitation was performed using the 1H6 antibody. The quantity and size of DNA fragments were verified on the Bioanalyzer with a "DNA 1000" chip. After precipitation, the DNA libraries were prepared using the "NEBext Fast DNA Library Prep Set for Ion Torrent" reference number E6270S. The libraries were then loaded on the "Ion Chef System" and sequenced on the Ion Torrent. Raw reads were aligned on the human genome Hg19 using the torrent platform. EaSeq software (V. 1.111)[86] (http://easeq.net) was used in order to analyze the aligned data, and to call the peaks. EaSeq was also used in order to produce the heatmaps and the train plots with a window of 10 kb. Annotation and the statistical significance were done using the RegionR package (V. 1.18.1)[87] using databases downloaded from UCSC table browser[88]. In order to identify the motifs, we used the MEME-ChIP from the MEME suite (V. 5.0.3)[89,90]. In order to identify the pathways of the identified genes, we used Gene Ontology (GO) (Panther 14.0)[91–93]. The ChIP-seq data of the 1H6 antibody performed in this publication have been deposited in NCBI's Gene Expression Omnibus (Edgar et al., 2002) and are accessible through GO Series accession number GSE133113.

**Transcriptome analysis**. Total RNA from two independent biological samples were extracted from each of the two cell line derived from AD patients along with the two cell lines derived from healthy patients for a total of eight samples, using the standard procedure of Qiagen columns and assayed for RNA integrity. cDNA was prepared according to the manufacturer's instructions (NEB library) and sequenced using the Illumina platform. Base-calling and feature count were done using Illumina software. For differential expression analysis, Dseq2[94] was used on R program[95]. For the gating strategy please refer to Supplementary Fig. 11a–c. The RNA-seq data were deposited and are accessible through GO series accession number GSE162873.

**Reporting summary**. Further information on research design is available in the Nature Research Reporting Summary linked to this article.

## Data availability
Public data accession numbers used in this study are: GSE22162, GSE44849, GSE76688, GSE38273. The ChIP-seq data of the 1H6 antibody performed in this publication have been deposited in NCBI's Gene Expression Omnibus and are accessible through GO Series accession number GSE133113. These data have been used to construct Supplementary Figs. 5–7. The RNA-seq data were deposited and are accessible through GO series accession number GSE162873. All data are also available from the authors upon reasonable request.

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

## Acknowledgements

This work was supported by grants from the National Science and Engineering Research Council of Canada (NSERC), Canadian Institutes of Health Research (CIHR), Maisonneuve-Rosemont Hospital Foundation, and Pierre Theroux Family Foundation for Alzheimer's Disease Research. A. F., R. H., and A. B. were supported by fellowships from the Molecular Biology Program of Université de Montréal.

## Author contributions

G. B. and R. H.: conceived and designed; R. H., A. F., and A. B.: performed the experiments; G. B., R. H., A. F., and A. B.: analyzed the data; G. B., R. H., A. F., and A. B.: wrote the paper.

## Competing interests

A. F. and G. B. are co-founders and shareholders of StemAxon™. The corporation was however not involved in this study. R. H. and A. B. declare no competing interests.
