## [Peer Review File · Nature Communications]

Reviewers' comments:

Reviewer #1 (Remarks to the Author):

Hanna et al. investigate the link between transcription, chromatin state and G4 DNA sequences in normal and Alzheimer diseased neurons. The authors suggest a link between BMI1 loss, chromatin relaxation and induction of G4 structures and show genomic preference for G4 hotspots around LINE1 elements. They finally demonstrate that suppression of RNA polymerase is sufficient to inhibit induction of most G4 structures.

While the overall quality of experiments appears rigid - most of my technical concerns relate to the computational analyses – I disagree on the author's implications:

It appears, G4 structures simply occur at sites of active transcription (which associate with open/relaxed chromatin), probably as a byproduct of RNA-polymerase mediated unwinding of DNA. So any observed differences between control/BMI1 knockdown, normal/diseased neurons, untreated/treated cells should be first analyzed through the prism of transcriptional de-regulation.

For example:

- Canonical lamina associated domains become more permissive for transcription in AD, hence enrichment of G4 compared to normal neurons.

- G4 coincide with LINE1 elements because they are transcriptionally active in those cells

This leads to another problem. Assuming intragenic LINE1 elements represent active promoters, they likely obscure the author's analyses with respect to gene transcript quantification and splicing isoforms. Any active LINE1 element might contribute to novel splice junctions and or higher exonic transcript levels, but this is independent of G4 structures who just happen to follow transcription.

So under the transcription-link assumption (which I'm happy to see disproven), I'm afraid the current study represents a list of true genomic correlates, associations, and enrichments which I disagree to sufficiently advance our knowledge about Alzheimer pathophysiology or regulatory impact of G4 structures.

In case the authors seek to revise their manuscript, I suggest to:

- Uncouple G4 structures from transcribed/untranscribed regions

- Investigate the differences between both groups (sequence, genomic context ...)

- Can they mutate G4 motifs in LINE/LTR consensus sequences and find a difference in promoter activity (e.g. luciferase assay)

Technical concerns:

Exclude peaks from cluster 2 and 4 (shown in 3a) and focus on the regions with true enrichment. Around half of all G4 peaks overlap with LINE, SINE, and LTRs simultaneously, probably owing to the large size of their peaks (kbs?) and genomic repeat density. Any random set of kb intervals will strongly overlap with LINES/SINES and LTRs. So I suggest to narrow/center the peaks and link it to the most likely repeat locus (based on G4 motif prediction, e.g.)

Figure 4d. Please show statistical enrichment together with effect size. Not only for LINES, but also for LTRs (I assume overlap with SINES will disappear when narrowing peaks).

Reviewer #2 (Remarks to the Author):

This manuscript characterizes the relationship between BMI1 deficiency, G-quadruplex (G4) secondary structure involving heterochromatin regions, endogenous retroviral L1 elements, and defective transcription in Alzheimer's disease (AD). BMI1 is a polycomb group protein enriched at, but not bound to, heterochromatin regions. Decreased neuronal BMI1 expression has been associated with AD. The authors demonstrate that BMI knockdown in human dermal fibroblasts and murine retinal neurons results in upregulated G4 formation, associated with reduced H3K9me3 and increased H3K9ac in heterochromatin regions. Functional assays investigating modulation of chromatin structures via GSK3b and p53 inhibition demonstrate that the G4 phenotype is rescued upon re-establishment of normal chromatin compaction. The authors then move to G4 ChIP-seq analysis in iPSC neurons derived from normal and AD brains. They find that G4 positioning is enriched in laminin-associated domain (LAD) heterochromatin regions and associated with Long Interspersed Element-1 (L1) sequences, which are thought to be transcriptionally active. Moreover, they show by RNAPol II inhibition that G4 formation at these sites are dependent on

active transcription. Finally, they demonstrate that functionally relevant, G4-associated transcripts in AD neurons are differentially expressed and spliced.

This manuscript presents intriguing novel biology that may have disease relevance. However, concerns detailed below should be addressed.

1) The authors report that the increased G4s seen in AD neurons are not associated with increased DNA damage. This result is somewhat at odds with abundant literature establishing strong connections between G4s and DNA damage. The authors should show this data (it is not currently shown). They should also verify their observation using ChIP for gamma-H2AX and/or 53BP1, drawing correlations with WRN/1H6/BG4 ChIP.

2) Given the reported specificity issues with the 1H6 antibody (for example Kazemier et al., *Nucleic Acids Res*, 2017), along with the largely transcriptional effects observed by the authors, it is plausible that R-loops may contribute to the biology described in this manuscript. The authors should confirm that R-loops are not involved in both the BMI and AD-associated G4 phenotypes reported herein. Antibody reagents for R-loops are commercially available. Additionally, RNase and DNAase control experiments should be performed to further bolster the case that the structures seen in IF experiments are exclusively DNA-based.

4) The GSK and p53 inhibitor studies are intriguing. What are the effects on H3K9me3 levels in BMI knockdown contexts? Additionally, what is the impact of GSK and p53 inhibition on the transcriptional profiles of AD and normal iPSC neurons? In both experimental contexts, are the transcriptional consequences of increased G4s reverted? Does HDAC inhibition phenocopy the transcriptional profile of AD neurons in normal neurons?

5) The latter half of the paper, featuring the ChIP-seq, L1 element, and RNA-seq analyses, is primarily executed in iPSC neuronal models extracted from non-isogenic hosts. Accordingly, the impact of BMI1 in this biology is less clear. Have efforts been made to bolster BMI expression or knock it down in AD and normal neurons, respectively? What are the effects on G4s and gene expression?

More minor concerns.

1) How were LADs annotated in this study? By my reading, LAD coordinates were obtained from publically available data. The precise positioning of LADs differs considerably by cell type. Was an appropriate cell type used in this case?

2) I am not sure how to read/interpret FIG. 3a. Additional explanation should be provided in both the text and figure legend, defining what is shown.

3) The following statement should really be referenced. "Since BG4 peaks were reported to be predominant at the promoter of actively transcribed genes (HERE), these results suggested that BMI1 peaks containing putative G4 DNA sequences are rarely present in actively transcribed regions."

4) The last paragraph mentions potential "broader implications" of heterochromatin relaxation in other neurodegenerative diseases, however this is very vague. This part can be expanded further or rephrased to include an example.

Reviewer #3 (Remarks to the Author):

The manuscript by Hanna and colleagues examines the relationship between the formation of G4 quadruplexes, chromatin state and binding by the Polycomb group protein BMI1 and how this may play a role in Alzheimer's disease. The study contains data from a very wide range of models and techniques, showing BMI1 plays a role in modification of G4-mediated heterochromatin modifications, includes data from post-mitotic retinal neurons and cone receptors in *Bmi1*^{-/-} mice

and used ChIP-seq analysis to examine G4 peaks in evolutionary conserved Long Interspersed Elements. However, the the study design is not robust or convincing when reporting data from AD models which forms the crux of the work and the focus of the title.

The rationale for the study as related to AD is not strong as there is little link between AD and progeroid diseases. Familial AD genes do not encode helicases or DNA damage and/or repair proteins. The wide range of models studied from human primary fibroblasts, post-mitotic retinal neurons and cone receptors, iPSC neurons and post-mortem brain appears unfocussed. A better rationale for the work needs to be presented.

Major comments: The major experimental weakness of the study is a lack of robust experimental design and numerous inconsistencies in the AD work.

1. The iPSC modelling work is unconvincing. What is the nature of the “two distinct AD cases” from whom iPSCs were obtained? Are they familial AD (fAD) cases (in which case, which genes are involved?) or sporadic (sAD) cases. For fAD the accepted standard would need there to be iPSC lines from three individuals, unless isogenic controls are used. For sAD the number of lines required is unknown, but would be at least 3 and likely many more. Worse still, the data seem to be presented very selectively from the lines used eg: Fig 2d seems to show data from Control 1 v AD1, Fig 2F shows data from Control 2 v AD2 and Fig 5g shows Control 1 v AD2.2. All data presented must be from 3 well-defined AD lines v 3 control lines.

2. Is the data in Fig 3C derived from only 1 AD line v 1 Control line? Such a one v one study would be insufficient as it gives no indication of the variability within Controls and within AD, which is required in order to assess the significance of the difference between Controls and AD. Fig 3 seems to focus on AD1, but this switches back to AD2 for Fig 5. Again, all data presented must be from 3 well-defined AD lines v 3 control lines.

3. How were the neurons differentiated, prepared and isolated? Gene expression in cortical neurons can be highly heterogeneous and vigorous culture QC is required (see Volpato et al Stem Cell Reports 2018). How was an efficient differentiation confirmed; 30 days is a very short time to prepare cortical neurons, often over 100 days is needed to obtain mature neurons? Were other AD-related phenotypes observed in the same cultures to provide compelling evidence that the neurons represented an AD model? How many differentiations were performed?

4. In the Methods section text and in Fig 2F the authors seem to use the phrases “human embryonic stem cells” and iPSCs interchangeably which is concerning. Please clarify.

5. The lack of clarity and robustness is then repeated in the post-mortem studies: How many AD and healthy control brains studied? Such a number would typically be at least five AD v five control. Were they fAD or sAD, what Braak stage patients were chosen to study and which brain regions were chosen? The text says “Using frozen brain sections from the hippocampus of elderly controls and AD cases” but the Fig 2b legend says frontal cortex.

August 10, 2020
 NCOMMS-19-42281-T

REBUTTAL LETTER

Reviewer #1 (Remarks to the Author):

G4 structures simply occur at sites of active transcription, probably as a byproduct of RNA-polymerase mediated unwinding of DNA. So, any observed differences between control/BMI1 knockdown, normal/diseased neurons, untreated/treated cells should be first analyzed through the prism of transcriptional deregulation. Assuming intragenic LINE1 elements represent active promoters, they likely obscure the author's analyses with respect to gene transcript quantification and splicing isoforms. Any active LINE1 element might contribute to novel splice junctions and or higher exonic transcript levels, but this is independent of G4 structures who just happen to follow transcription.

This is a very important point, we agree. In AD neurons, genes are generally upregulated when compared to control neurons (**Fig. S11a**), which is consistent with chromatin relaxation and gene de-repression. However, when all genes associated with a G4 structure in AD neurons are analyzed by RNA-seq in a Volcano plot (comparing control and AD neurons), there is no correlation between the presence of a G4 structure and differential gene expression. On the contrary, we observe a trend toward gene downregulation (**Fig. S11b**). Moreover, when genes with an intragenic G4 structure associated with abnormal splicing events are analyzed (**Fig. 5**), we observe that they are significantly downregulated in AD neurons (**Figs. 5 and S11c**). Thus, the formation of a G4 structure associated with L1 transcription results in aberrant splicing and reduced gene expression in AD neurons. Conversely, when a G4 structure is lost in AD neurons, such as the intergenic G4 structure lost between the SMN1 and NAIP locus (**Fig. 5**), SMN1 expression is increased and the transcript show extra-exon inclusion. These results do not fit the proposed interpretation.

Rather, our observations are consistent with previously published models, such as 1) Reduced gene expression owing to the stalling of RNA-Pol2 at a predicted intragenic G4 structure in ATRX-deficient cells; 2) Gene downregulation associated with exon exclusion in the process called *exon-mediated transcription starts*. These models are cited in the discussion of the manuscript.

Around half of all G4 peaks overlap with LINE, SINE, and LTRs simultaneously, probably owing to the large size of their peaks (kbs?) and genomic repeat density. Any random set of kb intervals will strongly overlap with LINEs/SINEs and LTRs. So I suggest to narrow/center the peaks and link it to the most likely repeat locus (based on G4 motif prediction, e.g.) Figure 4d. Please show statistical enrichment together with effect size. Not only for LINEs, but also for LTRs (I assume overlap with SINEs will disappear when narrowing peaks).

This is a very important point, we agree. This is why we have performed two additional independent analyses to test if the overlap observed between G4 peaks and repetitive elements is due or not to a random distribution:

- 1) The 1H6 peaks covered 0.15% (CTL) and 0.6% (AD) of the human genome, while predicted G4 sequences cover 0.3% of the genome. As shown in **Fig. 4a-c**, predicted G4 sequences have a significantly lower overlap with repeat elements than 1H6 peaks identified in the ChIP-seq experiment. Hence, 95% of 1H6 peaks overlapped with a LINE

sequence in control neurons (and 98% in AD neurons), while only 20% of predicted G4 sequences overlapped with a LINE sequence.

- 2) We further performed a permutation test with 1000 permutations of the 1H6 peaks (Bernart Gel et al. 2016, Bioinformatics). This means that for each chromosome, we have generated 1000 new sets of peaks that matched our original peaks in number and length. Please keep in mind that the randomized sets matched exactly the original peak set for what pertains to peak size distribution. These 1000 random peaks were then overlapped with repetitive elements. The results show a significantly different distribution than what has been obtained in the ChIP-seq experiment. The P-value of this test ($P < 0.001$) is shown in **Fig. 4 a-c**. We can see in **Fig. S7c** how the test works: here we compared the 1H6 peaks to the putative G4 sites, where we plotted the distribution of overlap between putative G4 sequences and the 1000 random permutations similar in size to the 1H6 peaks and the observed overlap pointed in a green line. This test removes any bias, linked to size distribution between the chromosomes and numbers of peaks, that can affect the overlap percentage.

Reviewer #2 (Remarks to the Author):

1) The authors report that the increased G4s seen in AD neurons are not associated with increased DNA damage. This result is somewhat at odds with abundant literature establishing strong connections between G4s and DNA damage. The authors should show this data (it is not currently shown). They should also verify their observation using ChIP for gamma-H2AX and/or 53BP1, drawing correlations with WRN/1H6/BG4 ChIP.:

We have investigated the correlation between G4 structures and DNA damage extensively: while AD neurons present numerous 53BP1 foci, the foci weakly co-localized with 1H6 foci (**Fig. S6a**). This weak colocalization is apparently specific to post-mitotic cells, such as neurons. Hence, we have observed that 1H6 does co-localize with large 53BP1 and gamma-H2Ax foci in *BMI1*-deficient human dermal fibroblasts (Hanna et al. In preparation). Thus, G4 structures are associated with the formation of DNA damage most preferentially in mitotically active cells.

2) Given the reported specificity issues with the 1H6 antibody (for example Kazemier et al., *Nuclei Acids Res*, 2017), along with the largely transcriptional effects observed by the authors, it is plausible that R-loops may contribute to the biology described in this manuscript. The authors should confirm that R-loops are not involved in both the BMI and AD-associated G4 phenotypes reported herein. Antibody reagents for R-loops are commercially available. Additionally, RNase and DNAase control experiments should be performed to further bolster the case that the structures seen in IF experiments are exclusively DNA-based.: We have investigated this in *BMI1*-deficient human dermal fibroblasts and found that while *BMI1* knockdown increased 1H6 staining (**Fig. S1c**), no significant increase in R-loops was observed using the DNA/RNA hybrid S9.6 antibody (**Fig. S1d**). S9.6 labeling was restricted to the nucleoli, as shown in the profile plot in **Fig. S1d**. We have also performed DNase and RNase control experiments: While DNaseI treatment was able to prevent 1H6 labeling (**Fig. S1e**), RNase H and RNase A treatments were unable to reduce the 1H6 staining (**Fig. S1f**), thus showing that the structures recognized by 1H6 are DNA-based structures and not RNA-DNA hybrids or RNA-based structures.

4) The GSK and p53 inhibitor studies are intriguing. What are the effects on H3K9me3 levels in BMI1 knockdown contexts? Additionally, what are the impact of GSK and p53 inhibition on the transcriptional profiles of AD and normal iPSC neurons? In both experimental contexts, are the transcriptional consequences of increased G4s reverted? Does HDAC inhibition phenocopy the transcriptional profile of AD neurons in normal neurons?: **Treatment of BMI1-knockdown neurons with GSK and p53 inhibitors rescues amyloid and p-Tau levels (Flamier et al. 2018, Cell Reports). It also has an impact on H3K9me3 levels (Flamier et al., In Preparation). Treatment of AD neurons with GSK3b or p53 inhibitors greatly rescues the chromatin relaxation phenotype (Figs. 4g and S5e), and apparently also greatly improve the differential gene expression profile (unpublished). HDACi treatment is not predicted to reproduce the transcriptional profile observed in AD neurons because it does not alter the same heterochromatin domains, but rather all of the chromatin. Thus, although the reviewer's questions are legitimate, we feel that they might be outside of the scope of the current manuscript, which is now very busy with 11 supplementary figures.**

5) The latter half of the paper, featuring the ChIP-seq, L1 element, and RNA-seq analyses, is primarily executed in iPSC neuronal models extracted from non-isogenic hosts. Accordingly, the impact of BMI1 in this biology is less clear. Have efforts been made to bolster BMI expression or knock it down in AD and normal neurons, respectively? What are the effects on G4s and gene expression?: **These are also very good and legitimate questions. Sporadic AD is not a monogenic disorder. Hence, isogenic hosts of patient-specific iPSC-derived AD neurons do not exist and cannot be generated. This is why we have used iPSC lines from 3 unrelated sporadic late-onset AD cases. Yet, all 3 cases show reduced BMI1 expression in differentiated neurons (Flamier et al. 2018, Cell Reports). The biology associated with BMI1 knockdown or overexpression in normal human neurons has been extensively studied and is published (Flamier et al. 2018, Cell Reports). We are now investigating the impact of BMI1 over-expression in AD neurons and this work is the object of another manuscript in preparation.**

More minor concerns.

1) How were LADs annotated in this study? By my reading, LAD coordinates were obtained from publically available data. The precise positioning of LADs differs considerably by cell type. Was an appropriate cell type used in this case? **We agree. Yet, to our knowledge, there is no description of LAD coordinates in human neurons. Thus, we have based our work on LADs using the article "Constitutive nuclear lamina–genome interactions are highly conserved and associated with A/T-rich sequence" by Meuleman and al. In this article, they analyzed the LADs of hES cells and HT1080 cells. When they found common LADs between these very distinct cell lines, they consider the said LADs as constitutive. All the other domains were considered as facultative.**

Reviewer #3 (Remarks to the Author):

The manuscript by Hanna and colleagues examines the relationship between the formation of G4 quadruplexes, chromatin state and binding by the Polycomb group protein BMI1 and how this may play a role in Alzheimer's disease. The study contains data from a very wide range of models and techniques, showing BMI1 plays a role in the modification of G4-mediated heterochromatin modifications, includes data from post-mitotic retinal neurons and cone receptors in Bmi1^{-/-} mice

and used ChIP-seq analysis to examine G4 peaks in evolutionary conserved Long Interspersed Elements. **However, the study design is not robust or convincing when reporting data from AD models which forms the crux of the work and the focus of the title.**

The rationale for the study as related to AD is not strong as there is little link between AD and progeroid diseases. Familial AD genes do not encode helicases or DNA damage and/or repair proteins. **As mentioned in the introduction of the manuscript, the study is not on familial AD, but on sporadic late-onset AD, for which disease-causing genetic mutations have not been found. Work from others and we suggest (as specified in the introduction) that neurons from late-onset sporadic AD patients display “aging hallmark features” such as laminopathy, loss of heterochromatin, de-repression of satellite repeats (i.e. repetitive sequences) and reduced BMI1 expression (a senescence-inhibitor gene). Bmi1^{+/-} mice also display progeroid features in the brain and eye (senescence-associated beta-galactosidase activity, loss of heterochromatin, activation of satellite repeats, and lens cataracts (El Hajjar et al. 2019). Very similar anomalies are found in fibroblasts from patients with the progeroid syndrome (Werner, HGP, etc.). Aging is also the most important risk factor to develop sporadic AD. Thus, the idea that sporadic late-onset AD is possibly a premature brain aging disorder is well accepted in the scientific community. Not surprisingly, fibroblasts from patients with Werner and HGP syndromes also present induction of G4 structures (Hanna et al. In preparation).**

The wide range of models studied from human primary fibroblasts, post-mitotic retinal neurons and cone receptors, iPSC neurons and post-mortem brain appears unfocused. A better rationale for the work needs to be presented. **Work using fibroblasts and Bmi1-null mice was used to dissect the molecular mechanism linking BMI1 deficiency and induction of G4 structures. It was also used to perform several control experiments essential to validate the antibodies (1H6 and BG4) and the experimental conditions used later to study G4 structures in iPSC-derived sporadic AD neurons and brains. We believe it is a strength of the study and that it should be recognized as such.**

Major comments: The major experimental weakness of the study is a lack of robust experimental design and numerous inconsistencies in the AD work.

1. The iPSC modelling work is unconvincing. What is the nature of the “two distinct AD cases” from whom iPSCs were obtained? Are they familial AD (fAD) cases (in which case, which genes are involved?) or sporadic (sAD) cases. For fAD the accepted standard would need there to be iPSC lines from three individuals, unless isogenic controls are used. For sAD the number of lines required is unknown, but would be at least 3 and likely many more. Worse still, the data seem to be presented very selectively from the lines used eg: Fig 2d seems to show data from Control 1 v AD1, Fig 2F shows data from Control 2 v AD2 and Fig 5g shows Control 1 v AD2.2. All data presented must be from 3 well-defined AD lines v 3 control lines. **We apologize for any confusion. The 2 control and 2 AD iPSC lines generated in my laboratory were previously described in Flamier et al. 2018, Cell Reports. We added the clinical information of these patients in the manuscript that are as follows:**

Ctrl1: iPSCs derived fibroblasts from Coriell Institute #AG04152, from a healthy 82-year old male patient with no family history of Alzheimer's disease.

Ctrl2: iPSCs derived fibroblasts from Coriell Institute #AG09602 from a healthy 92-year old female patient with no family history of Alzheimer's disease.

AD1: iPSCs derived fibroblasts from Coriell Institute #AG08243 from a 72-year old male patient diagnosed with sporadic Alzheimer disease with no family history of Alzheimer's disease.

AD2: iPSCs derived fibroblasts from Coriell Institute #AG08259 from a 90-year old male patient diagnosed with sporadic Alzheimer's disease at the age of 87 with no family history of Alzheimer disease.

We also have an unpublished manuscript showing the full characterization of these new iPSC lines. This manuscript can be provided for reviewer eyes only, if required. We have also investigated BMI1 expression, heterochromatin state, and G4 structures in sporadic AD neurons produced from the Goldstein laboratory (Israel et al. 2012, Nature). The clinical information of this case was also added to the manuscript as follow:

LG: iPSCs derived from fibroblasts from Coriell Institute #GM24666 from a 83-year old male patient diagnosed with sporadic Alzheimer disease at the age of 78 with no family history of Alzheimer's disease and with a 3/3 APOE genotype.

We have also amended our figures to include results obtained with neurons from these 3 AD iPSC lines (**Figs. 4g, S5, and S9**).

2. Is the data in Fig 3C derived from only 1 AD line v 1 Control line? Such a one v one study would be insufficient as it gives no indication of the variability within Controls and within AD, which is required in order to assess the significance of the difference between Controls and AD. Fig 3 seems to focus on AD1, but this switches back to AD2 for Fig 5. Again, all data presented must be from 3 well-defined AD lines v 3 control lines. **The entire genome-wide ChIP-seq experiments and analyses were performed on AD1 and CTL1 neurons, in duplicate, + the input for each group. A single ChIP-seq experiment of this kind is very expensive and has required ~18 months of analysis, only for the bioinformatics part. In Figure 5, RNA-seq analyses were performed in quadruplicate on AD1, AD2, CTL1, and CTL2 neurons.**

3. How were the neurons differentiated, prepared, and isolated? Gene expression in cortical neurons can be highly heterogeneous and vigorous culture QC is required (see Volpato et al Stem Cell Reports 2018). How was an efficient differentiation confirmed; 30 days is a very short time to prepare cortical neurons, often over 100 days is needed to obtain mature neurons? **The entire neuronal differentiation method was fully described in Flamier et al. 2018, Cell Reports. We again described the differentiation protocol in material and method and recapitulated it in a schematic presented in Fig. 2f. Differentiation was confirmed by multiple methods, including RNA-seq, Western blot, and immunofluorescence. While “neurons” were differentiated for 30 days, the entire neuronal differentiation procedures lasted in fact 54 days (as shown in Fig. 2f), when neurons are post-mitotic and express NeuN, FOXG1, Tuj1, and MAP2. The cultures were arrested at DIV30 because AD neurons undergo degeneration at this stage.**

Were other AD-related phenotypes observed in the same cultures to provide compelling evidence that the neurons represented an AD model? How many differentiations were performed? **Yes, they do show a robust AD-related phenotype, including elevated secretion of ab42, reduced dendritic**

length, and reduced BMI1 expression (Flamier et al. 2018, Cell Reports). They also show many other anomalies, such as p-Tau accumulation, apoptosis, and a laminopathy (Flamier et al. In preparation). For the work presented here, the number of independent cultures is about 8 for each iPSC line.

4. In the Methods section text and in Fig 2F the authors seem to use the phrases “human embryonic stem cells” and iPSCs interchangeably which is concerning. Please clarify: **We apologize for any confusion. This was corrected accordingly. Human ESCs were only used for the BMI1 knockdown experiment presented in Fig. 2a and the HDACi experiment in Fig. 4i. Otherwise, all other experiments involving pluripotent stem cells were performed using iPSCs.**

5. The lack of clarity and robustness is then repeated in the post-mortem studies: How many AD and healthy control brains studied? Such a number would typically be at least five AD v five control. Were they fAD or sAD, what Braak stage patients were chosen to study and which brain regions were chosen? The text says “Using frozen brain sections from the hippocampus of elderly controls and AD cases” but the Fig 2b legend says frontal cortex.: **We again apologize for any confusion. This was corrected accordingly. The frozen sections were from the hippocampus of end-stage disease (Braak stage V/VI) sporadic late-onset AD patients (n = 3) and non-demented age-match controls (n = 3). The samples’ sections were kindly provided by the Banner Health Institute. In the control section, there was an average of 0.75 positive nucleus/section (1H6 staining) by IHC, while in AD patients, there was an average of 12.6 positive nucleus/section with a P-value <0.0001. This information is now included in the Figure legends section (Figure 2b).**

REVIEWER COMMENTS

Reviewer #1 (Remarks to the Author):

The authors have addressed my primary concern.

Minor comment:

Rather than just showing statistical significance, include the results of the permutation experiment in Fig4a-c along the empirical data (showing perc overlap).

Reviewer #2 (Remarks to the Author):

I have no further issues with this manuscript.

Reviewer #3 (Remarks to the Author):

The main concern of this Reviewer were the apparent selective presentation of data from sporadic AD and control iPSC-derived neurons across the Figures, and the poor level of detail given describing the post-mortem studies.

The iPSC work has been much improved in the revised manuscript with the authors adding a third AD line to the study and adding extra data such that many experiments (eg: Fig 2F) now show data from three AD and two control lines. This is very much to be welcomed.

However, in places the manuscript remains opaque as to the details of which lines were used. This must be clarified. For example, the line "In order to localize G4 structures on the human genome, we performed ChIP-seq on control and AD neurons at DIV30..." should be explicit and state that one AD (AD1) and one control (Cntrl 1) iPSC line was used.

Also, it remains the case from the data shown that Fig 3 shows data from AD1 and Fig 5 shows data from AD2 which is unfortunate as it means a direct comparison can not be done. You can not compare ChIP-Seq on AD1 with RNA-Seq in AD2. No-where does it state how many AD and Control iPSC lines were used for the RNA-Seq. The sentence "comparative RNA-seq analysis of control and AD neurons revealed..." again should state how many lines were used. If it was two, as stated in the rebuttal letter, then does the data in the Fig 5b schematic apply to AD1 as well which would allow comparison with Fig 3? If RNA-Seq data exists for AD1 then data for AD1 must be presented in Fig 5 as well as AD2 and a meaningful comparison between RNA-Seq and ChIP-Seq in the same line can be done. In Fig 5d: are the two AD western blot lanes AD1 and AD2, and the Control lanes Cntrl 1 and Cntrl 2: the legend does not say? It would be good if it was and the legend can be modified for clarity. Are the volcano plots in 5e showing AD1 and AD2 combined compared to Cntrl 1 and Cntrl 2 combined? Again, it does not say. It is essential to provide absolute clarity for the reader as to where the data come from.

The information on the post-mortem studies has been much improved.

December 8, 2020
 NCOMMS-19-42281-T

REBUTTAL

LETTER

Reviewer #1 (Remarks to the Author): The authors have addressed my primary concern.

Minor comment: Rather than just showing statistical significance, include the results of the permutation experiment in Fig4a-c along the empirical data (showing perc overlap).

Every single column of the graph is associated with a graph. In our opinion, adding all these graphs (66 graphs) will overburden the manuscript. Instead, we are showing the most important results i.e. graphs linked to the analysis of LINE, L1 and ALU sequences in the new Supplementary Figure 8.

Reviewer #2 (Remarks to the Author): I have no further issues with this manuscript.

Reviewer #3 (Remarks to the Author): The main concerns of this Reviewer were the apparent selective presentation of data from sporadic AD and control iPSC-derived neurons across the Figures, and the poor level of detail given describing the post-mortem studies. The iPSC work has been much improved in the revised manuscript with the authors adding a third AD line to the study and adding extra data such that many experiments (eg: Fig 2F) now show data from three AD and two control lines. This is very much to be welcomed.

a) However, in places the manuscript remains opaque as to the details of which lines were used. This must be clarified. For example, the line “In order to localize G4 structures on the human genome, we performed ChIP-seq on control and AD neurons at DIV30...” should be explicit and state that one AD (AD1) and one control (Cntrl 1) iPSC line was used.

The ChIP-seq experiment was performed on Ctrl2 and AD1 neurons. This information is now included in the corresponding Figure legend.

b) Also, it remains the case from the data shown that Fig 3 shows data from AD1 and Fig 5 shows data from AD2 which is unfortunate as it means a direct comparison cannot be done. You cannot compare ChIP-Seq on AD1 with RNA-Seq in AD2. No-where does it state how many AD and Control iPSC lines were used for the RNA-Seq. The sentence “comparative RNA-seq analysis of control and AD neurons revealed...” again should state how many lines were used. If it was two, as stated in the rebuttal letter, then does the data in the Fig 5b schematic apply to AD1 as well which would allow comparison with Fig 3?

a) Accordingly, we have performed RNA-seq analysis of AD1 neurons. The results of this are now presented in Figures 5, S13.

b) Figure 5b is the comparison between Ctrl1 and Ctrl2 vs AD1 and AD2 samples.

c) Figure 5c and 5e is the comparison between Ctrl1 and Ctrl2 vs AD1 samples.

d) Figures S11b, c and S14 is the comparison between Ctrl1 and Ctrl2 vs AD2 samples.

c) If RNA-Seq data exists for AD1 then data for AD1 must be presented in Fig 5 as well as AD2 and a meaningful comparison between RNA-Seq and ChIP-Seq in the same line can be done.

Accordingly, the analysis for AD1 is now presented in Figure 5 while the analysis for AD2 is presented in Figure S11. Further analyses of the RNA-seq data are also presented in Figures S13-14.

d) In Fig 5d: are the two AD western blot lanes AD1 and AD2, and the Control lanes Cntrl 1 and Cntrl 2: the legend does not say? It would be good if it was and the legend can be modified for clarity.

The Western blots were performed using the Ctrl2 and AD2 samples. This information is now included in the corresponding Figure legend.

e) Are the volcano plots in 5e showing AD1 and AD2 combined compared to Cntrl 1 and Cntrl 2 combined? Again, it does not say. It is essential to provide absolute clarity for the reader as to where the data come from.

a) Volcano plots in Figure 5e are the comparison between Ctrl1 and Ctrl2 vs AD1.

b) Volcano plots in Figure S11c are the comparison between Ctrl1 and Ctrl2 vs AD2 samples. This information is now included in the respective Figures and Figure legends.

REVIEWERS' COMMENTS

Reviewer #3 (Remarks to the Author):

The authors have addressed the remaining concerns adequately.

I have no further comments.